



# Quantification of Ecohydrological Sensitivities and Their Influencing Factors at the Seasonal Scale

Yiping Hou[1], Mingfang Zhang[2,3], Xiaohua Wei[1], Shirong Liu[4], Qiang Li[5], Tijiu Cai[6], Wenfei Liu[7], Runqi Zhao[8], Xiangzhuo Liu[9]

[1]Department of Earth, Environmental and Geographic Sciences, University of British Columbia (Okanagan campus), 1177 Research Road, Kelowna, British Columbia, V1V 1V7, Canada
[2]School of Resources and Environment, University of Electronic Science and Technology of China, Chengdu, 611731, China
[3]Center for Information Geoscience, University of Electronic Science and Technology of China, Chengdu, 611731, China
[4]Research Institute of Forest Ecology, Environment and Protection, Chinese Academy of Forestry, Beijing, 100091, China
[5]Department of Civil Engineering, University of Victoria, Victoria, British Columbia, V8W 2Y2, Canada
[6]Department of Forestry, School of Forestry, Northeast Forestry University, Harbin, 150040, China
[7]Jiangxi Provincial Key Laboratory for Restoration of Degraded Ecosystems & Watershed Ecohydrology, Nanchang Institute of Technology, Nanchang, 330099, China
[8]Division of Ocean Science and Technology, Tsinghua Shenzhen International Graduate School, Tsinghua University, Shenzhen, 518055, China
[9]INRA, Centre INRA Bordeaux Aquitaine, URM1391 ISPA, 33140, Villenave d'Ornon, France

*Correspondence to*: Mingfang Zhang (mingfangzhang@uestc.edu.cn)

## Abstract

Ecohydrological sensitivity is defined as the response intensity of streamflow to per unit vegetation change. Understanding of ecohydrological sensitivity and its influencing factors is important for managing water supply, reducing water-related hazards and ensuring aquatic functions by vegetation management. However, this topic has rarely been examined. In this study, 14 large watersheds across various environmental gradients in China were selected to quantify ecohydrological sensitivities at the seasonal scale and to examine their influencing factors such as climate, vegetation, topography, soil and landscape. Based on the variables identified by correlation analysis and factor analysis, the prediction models of seasonal ecohydrological sensitivity were constructed to test their utilities for the design of watershed management and protection strategies. Our key findings were: (1) ecohydrological sensitivities were more sensitive in dry conditions than in wet conditions, for example, 1% LAI (leaf area index) change averagely induced 5.05% and 1.96% change in dry and wet season streamflows, respectively; (2) seasonal ecohydrological sensitivities were highly variable across the study watersheds with different climate condition, dominant soil type and hydrological regime; and (3) the dry season ecohydrological sensitivity was mostly determined by topography (slope, slope length, valley depth, downslope distance gradient), soil (topsoil organic carbon, topsoil bulk density) and vegetation (LAI), while the wet season ecohydrological sensitivity was mainly controlled by soil (topsoil available water holding capacity), landscape (edge density) and vegetation (leaf area index). Our study provided a useful and practical framework to assess and predict ecohydrological sensitivities at the seasonal scale. We expect that ecohydrological sensitivity prediction models can be applied to ungauged watersheds or watersheds with limited hydrological data to help decision makers




and watershed managers to effectively manage hydrological impacts through vegetation restoration programs. We conclude that ecohydrological sensitivities at the seasonal scale were varied by climate, vegetation and watershed property, and their understanding can greatly support management of hydrological risks and protection of aquatic functions.

## 1 Introduction

Natural rivers often have a distinctive seasonal pattern of flow, where flow is highly related to precipitation and shows large variations over dry and wet seasons. Seasonal flows determine ecosystem functions (Toledo-Aceves et al., 2011; Bruijnzeel et al., 2011; Salve et al., 2011), and their responses to vegetation change are highly variable and consequently affect watershed ecosystem equilibrium (Maeda et al., 2015). On the one hand, wet season flows and their variability regulate flood magnitudes (Arias et al., 2012), determine the structure of floodplains and channel morphology (Jansen and Nanson, 2010), and provide
opportunities of recruitment of large woody debris (Warfe et al., 2011; de Paula et al., 2011). On the other hand, dry season flows are critical for maintaining a stable water supply and protecting aquatic ecosystem, as well as playing important roles in sustaining aquatic biota and refuging juvenile fishes (Bunn et al., 2006; Palmer and Ruhi, 2019). However, seasonal streamflow can be significantly affected by forest or vegetation change (Dai, 2011; Hirabayashi et al., 2013). Researches have showed that vegetation change can influence water retention time (Moore and Wondzell, 2005; Baker and Wiley, 2009; Bisantino et al.,
2015), alter snow accumulation and snowmelt processes (Lin and Wei, 2008; Zhang and Wei, 2012; Calder, 2005), and route river flow quickly to downstream (Winkler et al., 2010; Chang, 2012) and consequently increase the frequency and size of floods in wet season. Vegetation change can also affect dry season flow, which may influence baseflow level and the risk of droughts, and degrade or enrich in-channel habitat for aquatic species (Simonit and Perrings, 2013; Sun et al., 2016). Thus, understanding of seasonal hydrological variations to vegetation change is critical for maintaining sustainable water supply,
preventing large floods and droughts, and developing sound watershed management plans.

      Obviously, seasonal streamflow response to vegetation change is highly variable among watersheds worldwide. To better understand the general pattern of streamflow response to vegetation change, a uniform indicator named ecohydrological sensitivity (defined as the response intensity of streamflow to per unit forest or vegetation change) has been firstly introduced by Zhang et al. (2017). And ecohydrological sensitivity is believed to be controlled not only by the proportion of forest or
vegetation cover change but also by climate condition, hydrological regime and forest or vegetation type (Zhang et al., 2017; Li et al., 2017). Assessing ecohydrological sensitivity can provide various benefits. For example, it provides a dimensionless index on vegetation-water relationship so that any watersheds can be effectively compared. It also allows for predicting ecohydrological sensitivities for a landscape or region so that negative hydrological impacts in the areas with high ecohydrological sensitivities can be minimized through suitable arrangements of vegetation or watershed management
strategies. However, in spite of its usefulness, ecohydrological sensitivity and its influencing factors have been rarely quantified. To our knowledge, there is no any study on quantification of seasonal ecohydrological sensitivity.



Ecohydrological sensitivity is likely varied with time scales. The hydrological responses to vegetation change at the annual scale are the averaged and cumulative effects from those at shorter time intervals, which are typically associated with total annual magnitudes such as water yield or production, while those at daily or monthly or seasonal scales affect flow patterns and are normally related to floods and droughts. The seasonal scale is a medium level between daily and annual scales, which can affect both magnitude and pattern in terms of hydrological response and sensitivity. For example, the interactions between vegetation and water are quite different between dry and wet seasons (Donohue et al., 2010; Asbjornsen et al., 2011). Abundant water is available for vegetation growth in wet season, while vegetation in dry season mostly relies on limited soil moisture or groundwater for limited photosynthesis and transpiration. Besides, streamflow generation in wet season is mainly based on precipitation or water input, whereas dry season flow is controlled by soil moisture in the antecedent wet season and groundwater discharge. Thus, the contrasted processes in different seasons suggest that ecohydrological sensitivity must be examined at a seasonal scale.

Various factors including climate, vegetation and watershed property affect hydrological responses or sensitivities (Zhou et al., 2015; Li et al., 2017; Zhang et al., 2017). For examples, hydrological responses to land cover change tend to be more sensitive in non-humid regions (Zhou et al., 2015). Evapotranspiration change related to vegetation change is controlled by energy and water (Zhang et al., 2004; Creed et al., 2014; Yang et al., 2007). Topography controls hydrological processes by affecting the distribution and routing of water (Woods, 2007). Soil and landscape conditions are important for erosion, sediment and flow connectivity (Borselli et al., 2008). Clearly, fully assessing and understanding ecohydrological sensitivity requires a consideration of various influencing variables. Although past studies have focused on the hydrological influences of a single type of variables such as vegetation change (Beck et al., 2013; Feng et al., 2016; van Dijk et al., 2012), climate (Creed et al., 2014; Miara et al., 2017), topography (Lyon et al., 2012; Jencso and McGlynn, 2011; Li et al., 2018a) and landscape (Nippgen et al., 2011; Buma and Livneh, 2017; Teutschbein et al., 2018), the inclusion of various types of variables into an integrated assessment framework of hydrological responses has been a challenging subject.

China has experienced substantial and dynamic vegetation change over the past few decades. Deforestation and biomass loss dominated vegetation change from 1950s to 1980s (Wei et al., 2008), while the large-scale revegetation programs have been implemented since 1980s (Li et al., 2018b). Substantial and dynamic vegetation change in China provides such a great opportunity for assessing seasonal ecohydrological sensitivity and its influencing factors. The objectives of this study were: (1) to evaluate seasonal ecohydrological sensitivity in the selected large watersheds across environmental gradients; (2) to examine the role of climate, vegetation, topography, soil and landscape in seasonal ecohydrological sensitivity; and (3) to simulate and predict seasonal ecohydrological sensitivity based on the selected factors.



## 2 Study watersheds and data

### 2.1 Study watersheds

Fourteen large watersheds across climatic zones with the area ranging from 832 to 19189 km$^2$ in China were selected in this study. They include the Pingjiang and Xiangshui watersheds in Southeast China, the Tangwang River and Xinancha River

watersheds in Northeast China, the Upper Zagunao, Zagunao, Upper Heishui River, Heishui River, Gongbujiangda and Gengzhang watersheds in Southwest China and the Dongchuan, Heishuichuan, Jingchuan and Rui River watersheds in Northwest China (Fig. 1). Table 1 provides a brief summarise of climate, vegetation and topography in the study watersheds. The selected watersheds are mainly located in subtropical monsoon climate, temperate continental monsoon climate, alpine climate and temperate continental climate zones. In addition, substantial vegetation restoration programs caused large-scale

vegetation change from 1980s onwards. To evaluate seasonal ecohydrological sensitivity, the study periods start from 1983. Detailed information about climate, topography, soil and vegetation in each watershed can be found in the Supplement (Sect. S1).

### 2.2 Data

Daily or monthly discharges for 14 watersheds were obtained from various government agencies. The details about the study

periods and hydrometric stations can be found in the Supplement (Table S3). Discharges (m$^3$/s) were converted into the unit of mm according to drainage area. A hydrological year was equally divided into dry season and wet season, and then seasonal flows were calculated accordingly.

      The historical climate data used in this study include three sources: daily climate records from National Meteorological Information Centre of China Meteorological Administration (CMA: http://data.cma.cn/), spatial-interpolated

gridded climate data by use of the ANUSPLIN model and meteorological data collected at the associated hydrological stations or rain gauges (Table S3). In this study, daily or monthly climate data including mean temperature ($T_{mean}$), minimum temperature ($T_{min}$), maximum temperature ($T_{max}$) and precipitation (P) were derived and calculated accordingly. Monthly potential evapotranspiration (PET) was calculated based on estimated $T_{max}$ and $T_{min}$ by using Hargreaves' equation (Hargreaves and Samani, 1985).

120       Moderate Resolution Imaging Spectroradiometer (MODIS) land cover product MODIS MCD12Q1 with spatial resolution of 500m were downloaded from Land Process Distributed Active Archive Centre (LPDAAC: https://lpdaac.usgs.gov/products/mcd12q1v006/) (Sulla-Menashe et al., 2019). Because MODIS MCD12Q1 has 17 types of land covers, we reclassified all land cover types into forest, shrubland, grassland, agricultural, snow and other lands. MODIS MCD12Q1 in the year of 2001 was used to derive forest coverage and vegetation coverage (total coverage of forest, shrubland

and grassland) (Table S2).

      Leaf area index (LAI) derived from the Global Land Surface Satellite LAI Product (GLASS LAI) was used as a vegetation index to express vegetation change in this study (GLASS: http://glass-product.bnu.edu.cn/). The GLASS LAI



product dataset provides continuous global LAI at a high temporal resolution of eight days (Liang et al., 2013; Xiao et al., 2014). There are two types of GLASS LAI products with different spatial resolutions and available periods. The first GLASS

LAI product is based on Advanced Very High Resolution Radiometer (AVHRR) reflectance data with spatial resolution of 0.05°, and this dataset is available from 1982 to 2016. The other one, with a higher spatial resolution of 1 km is retrieved from Moderate Resolution Imaging Spectroradiometer (MODIS) reflectance data, but it only covers a period of 17 years from 2000 to 2016. As the study watersheds are large (>500 km$^2$) and the study periods are ended before 2006, the former GLASS LAI product was chosen for this study.

Harmonized World Soil Database (HWSD) published by Food and Agriculture Organization (FAO) and International Institute for Applied Systems Analysis (IIASA) with the spatial resolution of 1km was used to collect soil indices (Wieder, 2014). HWSD classifies soil into topsoil from surface to 30 cm below ground, and subsoil between 30 cm and 100 cm below ground.

      DEMs with the spatial resolution of 30m derived from GDEMDEM were provided by Geospatial Data Cloud site,
Computer Network Information Centre, Chinese Academy of Sciences (http://www.gscloud.cn). Topographical information of the study watersheds was derived from DEMs.

## 3 Methods

### 3.1 Definition and calculation of ecohydrological sensitivity

Zhang et al. (2017) introduced the ecohydrological sensitivity as the response intensity of annual streamflow to forest cover
change. In this study, we defined seasonal ecohydrological sensitivity ($S_f$) as the response intensity of seasonal streamflow variations to per unit vegetation change (using the leaf area index (LAI) as a proxy), which can be computed with equations (1)-(2). Seasonal streamflow variations to vegetation change ($\Delta Q_v$) were determined by an improved single watershed approach (see Sect. 2 in the Supplement for more details) (Hou et al., 2018a; Hou et al., 2018b). The value of seasonal ecohydrological sensitivity refers to the percentage of seasonal streamflow changes induced by 1% of LAI change. Ecohydrological sensitivity
is highly dependent on factors such as climate, hydrological regime, vegetation type, soil condition, and etc.

$$\Delta Q_v \% = 100 \times \frac{\Delta Q_v}{\bar{Q}} \tag{1}$$

$$S_f = \left| \frac{\Delta Q_v \%}{\Delta LAI} \right| \tag{2}$$

where, $\bar{Q}$ refers to the long-term mean seasonal streamflow during the study period; $\Delta Q_v$ is seasonal streamflow response to vegetation change in mm, which is calculated by the improved single watershed approach; $\Delta Q_v$% is seasonal streamflow
response to vegetation change in percentage (%); and $\Delta LAI$ is LAI variation compared to average LAI in the reference period in %.



### 3.2 Quantification of influencing drivers to seasonal ecohydrological sensitivities

Five types of indices including climate, vegetation, topography, soil and landscape were adopted in this study. Detailed information on the interpretations and calculations of 40 indices were presented in Table 2. Climate indices, including dryness

index and effective precipitation can demonstrate water input and climate condition in a given watershed (van Dijk et al., 2012; Jones et al., 2012; Zhang et al., 2004). Dryness index is calculated at the annual scale to demonstrate dryness condition, while effective precipitation (an integrated index of climatic variability) in dry season and wet season denotes seasonal water inputs. Vegetation growth is highly dependent on temperature, water, soil and geographical location (Chang, 2012). Vegetation coverage or forest coverage indicates a proportion of vegetation or forest in a watershed, but it cannot express vegetation

growth, mortality, and seasonality. Thus, LAI is recognized as a better indicator mainly because it is an important biophysical variable relating to photosynthesis, transpiration and energy balance (Launiainen et al., 2016; Verrelst et al., 2016; González-Sanpedro et al., 2008). Topographic indices can be classified into two groups: primary and secondary (also known as compounded topographic indices) (Li et al., 2018a; Moore et al., 1991). Primary topographic indices can be directly derived from DEM, whilst compounded topographic indices are based on one or more primary indices (Li et al., 2018a). In this study,

17 topographic indices including 5 primary indices and 12 compounded indices were selected to describe watershed characteristics like visibility, generation process and morphology (Yokoyama et al., 2002; Park et al., 2001; Jenness, 2004). Calculations of the topographic indices were made in ArcGIS 10.2 (ERSI) and SAGA GIS 2.1. Soil types were based on the FAO-85 system classification, while soil organic carbon and sanity were directly derived from HWSD in ArcGIS 10.2 (ERSI), and soil available water holding capacity, saturated hydraulic conductivity and bulk density were calculated using Soil-Plant-

Air-Water (SPAW) hydrology model. We used weighted average value to represent watershed-scale soil indices. Seven landscape indices including patch number (PN), patch density (PD), largest patch index (LPI), edge density (ED), contagion index (CONTAG), Shannon's diversity index (SHDI) and Simpson's diversity index (SIDI) at the landscape level were derived from FRAGSTATS 4.2 software.

According to dryness index (*DI*), watersheds were grouped into energy-limited (EL), equitant (EQ) and water-limited

(WL) conditions (McVicar et al., 2012). The most widely distributed soil type in a watershed was treated as the dominant soil type. Following our analysis, four dominant soil types (LIXISOLS, LUVISOLS, LEPTOSOLS and CAMBISOLS) were shown in this study. Additionally, the selected watersheds were categorized into rain-dominated (RD) and rain-snow hybrid (Hybrid) watersheds according to their hydrological regimes. Table 3 showed the detailed classifications for each watershed in terms of climate condition, dominant soil type and hydrological regime.

Non-parametric Mann Whitney U test was performed to detect the statistically significant differences between the watershed groups. Mann Whitney U test was applied to test whether there are significant differences in the median values of seasonal ecohydrological sensitivities between two groups (Birnbaum, 1956). Kendall correlation analysis and linear regression were used to identify statistically significant correlations between seasonal ecohydrological sensitivities and 40 indices at a significant level of *p*=0.10. The insignificant indices were excluded for prediction described below.



## 3.3 Prediction of seasonal ecohydrological sensitivity

Seasonal ecohydrological sensitivity can be predicted and simulated based on those identified significant variables. To accomplish this, factor analysis (FA) was introduced to further reduce the redundancy of indices. Similar to principal component analysis (PCA), indices after filtering by factor analysis could retain important information, which means that less indices can be used to represent most information (Lyon et al., 2012). Three criteria were used to pick highly related indices: the coefficient of Kaiser-Meyer-Olkin (KMO) test, $p$ value of Bartlett's test, and the diagonal coefficients of anti-image correlation matrix (Li et al., 2018a). Indices filtered by factor analysis with coefficient of KMO being greater than 0.50, the $p$ value of Bartlett's test being less than 0.05 and the diagonal coefficients of anti-image correlation matrix being greater than 0.50 were selected for further analysis.

Multiple linear regression model modified by stepwise regression was employed to predict seasonal ecohydrological sensitivity. Influencing factors filtered by correlation analysis and factor analysis were regarded as independent variables and ecohydrological sensitivity was considered as a dependent variable in a linear regression model. Independent variables were inputted into a model one by one, and ANOVA test was conducted accordingly. Once the $p$ value of ANOVA test was greater than 0.1, the input independent variable at this stage would be dropped. The optimal linear regression model was reached when no independent variables were inputted and no variables were dropped. The Akaike Information Criterion (AIC) and $R^2$ were used to find optimal multiple linear regression model for prediction. Except for quantitative indices, climate condition, dominant soil type and hydrological regime might also make contributions to prediction of ecohydrological sensitivity. As a result, we introduced dummy variables to quantify the influence of climate condition, dominant soil type and hydrological regime on model accuracy (Hardy, 1993). In this study, ecohydrological sensitivity based on the improved single watershed approach was called the observed $S_f$, while ecohydrological sensitivity from the multiple linear regression model was named as the predicted $S_f$.

## 4 Results

### 4.1 Seasonal ecohydrological sensitivity and its variations

Table 4 showed the comparison of ecohydrological sensitivities between the dry and wet seasons. The ecohydrological sensitivities in the dry season were significantly greater than those in the wet season (Fig. 2 and Fig. S8-S10). As shown in Fig. 2, 1% LAI change averagely induced 5.05% change in dry season streamflow, while in wet season, this value dropped to 1.96%. There were large variations in seasonal ecohydrological sensitivity among the study watersheds. The dry season ecohydrological sensitivity of the Tangwang River watershed was highest, up to 27.75, while the dry season ecohydrological sensitivity of the Upper Heishui River watershed was the lowest (1.01). Similarly, the wet season ecohydrological sensitivity with the value of 4.36 in the Tangwang River watershed was also the highest among all watersheds in the wet season, whereas the lowest wet season ecohydrological sensitivity (0.40) was found in the Xiangshui watershed (Table S4).





Comparisons of seasonal ecohydrological sensitivities were made among the study watersheds grouped by their climate conditions, dominant soil types and hydrological regimes (Fig. 3, Fig. 4 and Fig. 5). As suggested by Fig. 3 and Table 5, significant differences in both dry season and wet season ecohydrological sensitivities between energy-limited (EL) and equitant (EQ) watersheds and between energy-limited and water-limited (WL) watersheds were found. Significant differences

in the medians of wet season ecohydrological sensitivity in the pair of EQ-WL were also detected. 1% vegetation change caused 2.09%, 5.86% and 5.23% change of dry season streamflow in the energy-limited, equitant and water-limited watersheds, respectively (Fig. 3a), while it only led to 0.59%, 2.82% and 1.64% change of wet season streamflow in the EL, EQ and WL watersheds, respectively (Fig. 3b). These results clearly demonstrated that ecohydrological sensitivity was greater in the EQ and WL conditions, particularly in the dry season.

When seasonal ecohydrological sensitivity in watersheds grouped by dominant soil types was compared (Fig.4 and Table 5), the median of dry season ecohydrological sensitivity in the LIXISOLS-dominated watersheds was significantly different from those of the LUVISOLS- and CAMBISOLS-dominated watersheds at $\alpha$=0.05, and the significant differences in median of dry season ecohydrological sensitivity were also detected in the LUVISOLS-LEPTOSOLS, LIXISOLS-LEPTOSOLS, LUVISOLS-CAMBISOLS and LEPTOSOLS-CAMBISOLS pairs at $\alpha$=0.05 (Table 5). Similarly, the median

of dry season ecohydrological sensitivity in the LIXISOLS-dominated watersheds was significantly different from those of the LUVISOLS-, LEPTOSOLS- and CAMBISOLS-dominated watersheds at $\alpha$=0.05. On average 1% change in vegetation led to 2.11%, 3.29%, 5.62% and 13.01% change of dry season streamflow in the LIXISOLS-, LEPTOSOLS-, CAMBISOLS- and LUVISOLS-dominated watersheds, respectively (Fig. 4a), while it caused only 0.58%, 2.20%, 2.11% and 2.24% change of wet season streamflow (Fig. 4b).

Fig. 5 demonstrated the differences of seasonal ecohydrological sensitivity in watersheds grouped by hydrological regime. Mann-Whitney U test showed that there were significant differences between rain-dominated and hybrid watersheds in medians of dry season ecohydrological sensitivity (Table 5). 1% vegetation change can result in 6.51% and 3.29% change of dry season streamflow in rain-dominated and hybrid watersheds, respectively (Fig. 5a), while it only led to 1.75% and 2.20% change of wet season streamflow in rain-dominated and hybrid watersheds, respectively (Fig. 5b).

### 4.2 Prediction models for seasonal ecohydrological sensitivity

According to correlations between seasonal ecohydrological sensitivity and 40 indices detected by Kendall correlation and linear regression, 17 indices being significantly related to dry season ecohydrological sensitivity were identified (Table 6). Dry season ecohydrological sensitivity is significantly and positively correlated with dryness index (*DI*), topographic wetness index (TWI), downslope distance gradient (DDG), topographic positive openness (PO), topographic negative openness (NO), topsoil

salinity ($T_{ece}$), topsoil bulk density ($T_d$), while its correlations with all vegetation indices (LAI, vegetation coverage and forest coverage), slope, slope length factor (LS), terrain ruggedness index (TRI), valley depth (Depth), topsoil organic carbon ($T_{oc}$), patch density (PD) and edge density (ED) were significantly negative. In contrast, only 8 indices were significantly correlated with wet season ecohydrological sensitivity. Wet season ecohydrological sensitivity has a significantly positive correlation





with convergence (CON), topsoil available water holding capacity ($T_w$), topsoil saturated hydraulic conductivity ($T_{hy}$), subsoil
saturated hydraulic conductivity ($S_{hy}$), and subsoil salinity ($S_{ece}$) whereas a negative relation with effective precipitation ($P_e$),
soil types and edge density (ED).

8 out of 17 indices significantly related to dry season ecohydrological sensitivity was further identified by factor
analysis, which included factors such as DI, slope, LS, TWI, DDG, TRI, Depth and NO. For the factor analysis of dry season
ecohydrological sensitivity, the coefficient of KMO was 0.730, $p$ value of Bartlett's test was less than 0.05, and diagonal
coefficients of anti-image correlation matrix were greater than 0.53 (Table 7). Meanwhile, factor analysis identified 6 indices
($P_e$, CON, $T_w$, $T_{hy}$, $S_{hy}$ and ED) associated with wet season ecohydrological sensitivity based on correlation analysis. For wet
season subset, the coefficient of KMO with the value of 0.634 was lower than that in dry season subset, but diagonal
coefficients of anti-image correlation matrix were higher than those in wet season subset (≥0.57). The $p$ value of Bartlett's test
was 0.00. Given it is an important ecohydrological indicator for vegetation status in a watershed, LAI was also manually added
as a predictor in the predicted model. Fig. 6 showed the structure, parameters and statistics of the established prediction models
for ecohydrological sensitivity. The dry season model had a better performance with a higher $R^2$ of 0.966 (Fig. 6a), while the
$R^2$ was only 0.501 for the wet season model (Fig. 6b).

## 5 Discussion

### 5.1 Seasonal ecohydrological sensitivity and climate conditions

Climate conditions in terms of energy (temperature) and water (precipitation) are the most important drivers for vegetation
growth. Ecohydrological processes of vegetative watersheds vary greatly with climate conditions (Donohue et al., 2010). As
suggested by our study, both dry season and wet season ecohydrological sensitivities of the water-limited watersheds were
higher than those of the energy-limited watersheds (Fig. 3), and the dry season ecohydrological sensitivities were much higher
than the wet season ecohydrological sensitivities (Fig. 2). In addition, the dry season ecohydrological sensitivity significantly
increased with rising dryness index while the wet season ecohydrological sensitivity significantly decreased with increasing
effective precipitation (Table 6). In other words, under dry conditions (during dry periods or in dry regions), streamflow is
more sensitive to vegetation change than under wet conditions (during wet periods or in wet regions). These findings are in
accordance with results from previous studies, which indicate streamflow response to vegetation in drier regions might be
more pronounced than in wetter regions (Jackson et al., 2005; Vose et al., 2011; Li et al., 2017; Zhang et al., 2017). For
example, Farley et al. (2005) demonstrated that afforestation produced 27% water yield reduction in wetter sites, whilst 62%
water yield reduction was identified in drier sites based on the analysis of 26 catchments globally. Sun et al. (2006) modelled
streamflow responses to large-scale reforestation in China, and found increased vegetation cover produced a nearly 30%
reduction in streamflow in humid regions, but the streamflow reduction rose to approximately 50% in semi-arid and arid areas.
Creed et al. (2014) indicated water use efficiencies in forests were higher in drier years than in wetter years by assessing water
yield variations in North America. The different ecohydrological sensitivities between dry and wet seasons might be explained





by their various mechanisms of water use by vegetation. Vegetation growth in wet conditions with abundant available water, sufficient soil moisture and saturated aquifers is more sensitive to energy factors including temperature, radiation and heat input (Newman et al., 2006; Hou et al., 2018a; Zhang et al., 2011; Brooks et al., 2012). Changes in energy input in wet conditions can alter stomatal conductance and transpiration, and consequently affect the photosynthesis, transpiration and

biomass of vegetation (de Sarrau et al., 2012; Van Dover and Lutz, 2004). In contrast, in dry conditions with limited precipitation input, water is more critical for vegetation growth where vegetation mainly relies on its access to soil water through root systems to support photosynthesis and transpiration (Zhou et al., 2015).

### 5.2 Seasonal ecohydrological sensitivity and soil

Soils as the interface between streamflow and groundwater play vital roles in water cycle (Bockheim and Gennadiyev, 2010;

Schoonover and Crim, 2015). Our study showed that watersheds with different dominant soil types could have contrasting seasonal ecohydrological sensitivity. As shown in Fig. 4, the ecohydrological sensitivities in both dry and wet seasons in the LIXISOLS-dominated watersheds were the lowest as compared with those of CAMBISOLS-, LEPTOSOLS- and LUVISOLS-dominated watersheds. This result clearly illustrates the importance of soil types in hydrological responses and sensitivities (Rieu and Sposito, 1991; Srivastava et al., 2010; Chadli, 2016). Soil properties including organic carbon, salinity, available

water holding capacity, saturated hydraulic conductivity and bulk density can affect soil water infiltration and lateral movement (Hillel, 1974; Leu et al., 2010). For example, soil with higher available water holding capacity has the ability to store more water for vegetation growth (Mukundan et al., 2010). Saturated hydraulic conductivity is positively correlated to available water holding capacity, suggesting that soils in a watershed with higher value of saturated hydraulic conductivity might promote interactions between streamflow and groundwater (Sulis et al., 2010). Large differences between topsoil and subsoil

bulk densities suggest a frequent moisture movement, leading to more active interactions and feedbacks above and below soil (Zhao et al., 2010). LIXISOLS is characterized by lowest saturated hydraulic conductivity and smallest difference between topsoil and subsoil bulk densities as compared to other three types of soils (Table S1), indicating its lowest water storage capacity and less frequent water movement between topsoil and subsoil. Therefore, hydrological responses in the LIXISOLS-dominated watersheds were less sensitive to vegetation change, and consequently led to lowest seasonal ecohydrological

sensitivity.

### 5.3 Seasonal ecohydrological sensitivity and hydrological regimes

Hydrological regime is another determinant for ecohydrological sensitivity (Zhang et al., 2017). Our study found that the dry season ecohydrological sensitivity in the rain-dominated watersheds was significantly higher than that in the hybrid watersheds (Fig. 5), while insignificant difference in wet season ecohydrological sensitivity between them (Table 5). The differences in

dry season ecohydrological sensitivity between the rain-dominated and hybrid watersheds can be associated with their differences in the mechanisms of streamflow generation. In the rain-dominated watersheds, dry season streamflow is mainly maintained by groundwater discharge while both groundwater and snow water might be the sources of dry season streamflow



in the hybrid watersheds. Thus, the generation of the dry season streamflow in the hybrid watersheds tend to be more complex and stable, and can be more resilient to vegetation change in comparison with that in rain-dominated watersheds. This is
supported by several reviews which found that forest cover change in rain-dominated watersheds can produce greater hydrological impacts than in snow-dominated watersheds (Zhang et al., 2017; Moore and Wondzell, 2005). In hybrid watersheds, forestation or vegetation removal can lead to changes in snowmelt processes by altering snow accumulation, melting timing, energy input and wind speed in dry season (Frank et al., 2015), resulting in hydrological de-synchronization effects. These de-synchronization effects may offset negative impacts of vegetation change on dry season streamflow, and
eventually lower dry season ecohydrological sensitivity in the hybrid watersheds.

The lack of a significant difference in the wet season ecohydrological sensitivity between the rain-dominated and hybrid watersheds might be due to the fact that only precipitation form during wet season is rainfall. It is expected that there are similar interactions and feedback mechanisms between vegetation and water in wet season in all watersheds, leading to insignificant differences in wet season ecohydrological sensitivity between the rain-dominated and hybrid watersheds.

**5.4 Seasonal ecohydrological sensitivity and topography**

Topography as an dominating factor for hydrological processes (Zeng et al., 2016; Jenness, 2004; Scown et al., 2015; Yokoyama et al., 2002; Park et al., 2001; Li et al., 2018a) plays an important role in determining streamflow response to vegetation change (Price, 2011; Smakhtin, 2001). According to the established prediction model of dry season ecohydrological sensitivity (Fig. 6a), topographic factors including slope and downslope distance gradient had positive effects on dry season
ecohydrological sensitivity, while slope length factor and valley depth yielded negative effects. The vegetated watersheds with steeper slopes often have faster water movement from slopes to river channel and severe soil erosion in wet season if vegetation cover is destroyed, which can greatly reduce wet season soil water storage for supply to dry season streamflow, and therefore have greater dry season ecohydrological sensitivity (Desmet and Govers, 1996; Zhang et al., 2012). Similarly, vegetated watersheds with smaller slope length factor and valley depth can also have greater dry season ecohydrological sensitivity. This
is possibly because these watersheds generally have a generally flatter topography and longer water residence time, and consequently allow for more interactions between vegetation and water, which likely lead to greater ecohydrological sensitivity in dry season.

Unlike the dry season ecohydrological sensitivity, no topographic indices were associated with wet season ecohydrological sensitivity (Fig. 6b). As we know, climate and vegetation are two major drivers to hydrological variations in
forested watersheds (Wei et al., 2018; Li et al., 2017). This indicates that in wet season, climate plays a more dominate role in hydrological responses or variations, which means a decreasing role of vegetation on streamflow and consequently reduction of ecohydrological sensitivity. The decreasing role of vegetation on streamfow in wet season may explain the insignificant impact of topographic indices on wet season ecohydrological sensitivity.



## 5.5 Seasonal ecohydrological sensitivity and landscape

Landscape pattern can directly affect hydrological connectivity within a watershed, and can also indirectly influence hydrological processes by controlling soil activities such as soil erosion and sediment (Buma and Livneh, 2017; Teutschbein et al., 2018; Karlsen et al., 2016). Based on the prediction models of seasonal ecohydrological sensitivity, landscape pattern played a more important role in wet season ecohydrological sensitivity than in dry season ecohydrological sensitivity. Only edge density was identified as an effective, negative landscape predictor for wet season ecohydrological sensitivity.

Watersheds with a higher value of edge density are often featured by landscape fragmentation and segmentation, e.g., scatter distributed vegetation, higher road densities, leading to poor hydrological connectivity and high risk of soil erosion. The increasing role of watershed property (edge density) means that the relative role of vegetaion in hydrological reponse would be lower, which consequently lead to decreasing of wet season ecohydrological sensitivity.

## 5.6 Implications

Ecohydrological studies at the seasonal scale are limited due to the lack of the understanding of complex and variable streamflow responses to climate, vegetation, topography, soil and landscape (McDonnell et al., 2018; Singh et al., 2014; Wei et al., 2018; Li et al., 2018a; Oppel and Schumann, 2020; Guswa et al., 2020). Our findings clearly showed that seasonal ecohydrological sensitivity was not only highly associated with climate and vegetation change, but also significantly related to watershed properties like topography, soil and landscape. As indicated by the constructed prediction models, the dry season

ecohydrological sensitivity could be better described by vegetation, topography and soil (Fig. 6a) while the wet season hydrological response was mainly controlled by vegetation (leaf area index), soil (topsoil available water holding capacity) and landscape (edge density) (Fig. 6b). Given complex and variable hydrological responses to vegetation change among the study watersheds due to their differences in watershed properties (Zhou et al., 2015; Wei et al., 2018), our seasonal ecohydrological sensitivity prediction model can provide valuable information for understanding of the relative role of climate,

vegetation and watershed characteristics of topography, soil and landscape in seasonal ecohydrological processes (Fig. 6).

Since many watersheds lack long-term monitoring data on climate, hydrology and vegetation, a quantitative assessment of hydrological response to vegetation change at the watershed scale is very challenging and time-consuming. However, physical watershed data on climate, vegetation, watershed property can be easily derived from on-line climate datasets, remote sensing-based products, DEMs and soil databases. Development of a seasonal ecohydrological sensitivity

prediction model can be an efficient tool for watershed managers to evaluate hydrological impact of vegetation restoration programs with easily accessible data on climate, vegetation, topography, soil and landscape. Once seasonal ecohydrological sensitivity for different watersheds can be predicted quickly, future forest management can be implemented in a more sustainable way. We expect that the assessment framework from this study can be effectively applied to any watersheds where physical watershed data are available to support sustainable watershed planning and management.



## 6 Conclusions

Ecohydrological sensitivities at the seasonal scale were quantified in 14 large watersheds across various environmental gradients in China. Our main conclusions are: (1) hydrological responses were greater and more sensitive in dry conditions than in wet conditions; (2) seasonal ecohydrological sensitivities were highly variable across climate gradient, dominant soil type and hydrological regime; and (3) dry season ecohydrological sensitivity could be better controlled by vegetation, topography and soil while wet season hydrological sensitivity by vegetation, soil and landscape. Our study also demonstrated usefulness of constructing an ecohydrological sensitivity prediction model for predicting ecohydrological sensitivity in ungauged watersheds or watersheds with insufficient hydrological data to help watershed managers to effectively manage hydrological impacts and risks through vegetation restoration programs.

*Data availability.* Climate, vegetation, topography, soil and landscape indices of study watersheds are freely available upon request by sending an email to the corresponding author.

*Author contributions.* YH and MF proposed the analysis, designed the experiment, performed the result analysis and wrote the paper. QL and XW interpreted results and reviewed the manuscript. SL, TC, WL and XL collected the data. RZ calculated landscape indices. All authors participated in the 'Discussion' section.

*Competing interests.* The authors declare that they have no conflict of interest.

*Financial support.* This paper was supported by China National Science Foundation (No.31770759) and National Program on Key Research and Development Project of China (No. 2017YFC0505006).

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





**Figures:**

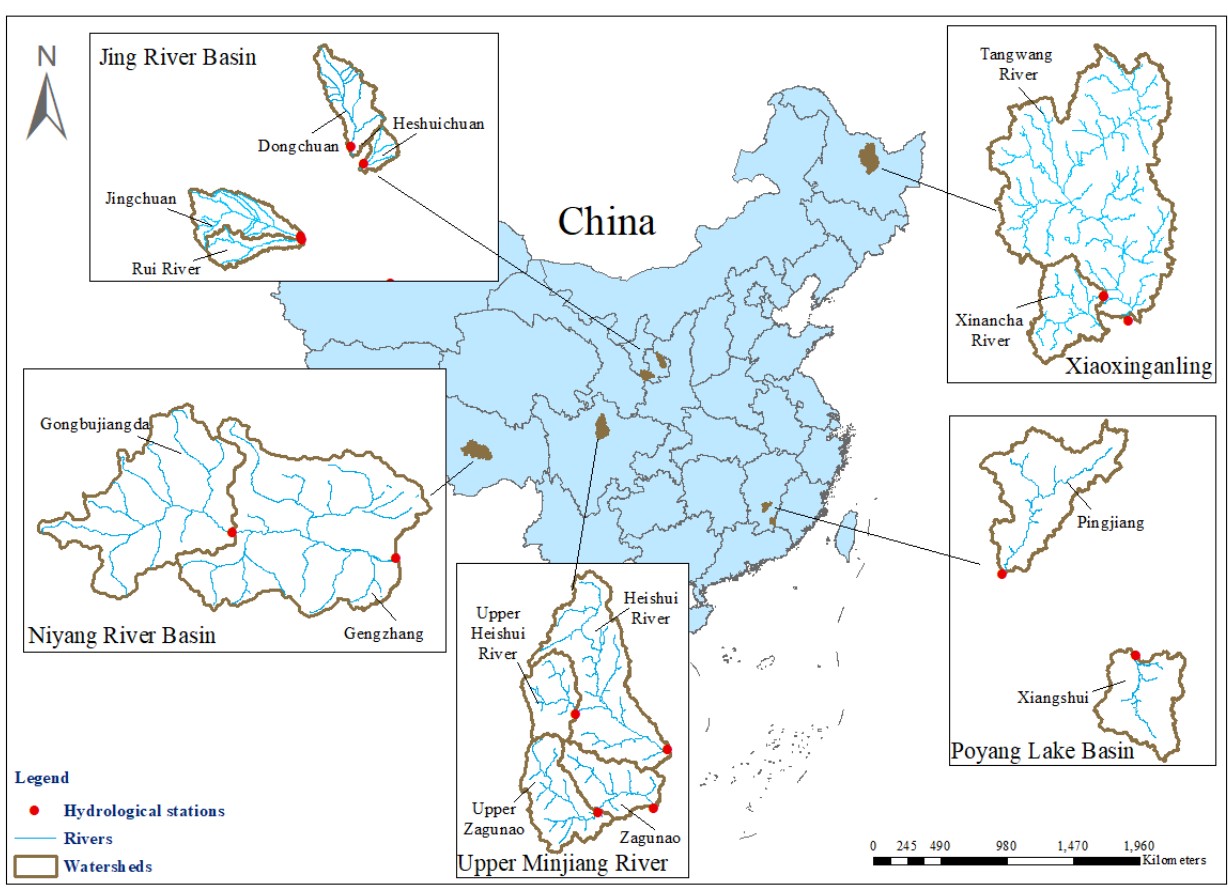


**Figure 1: Locations of the study watersheds.**





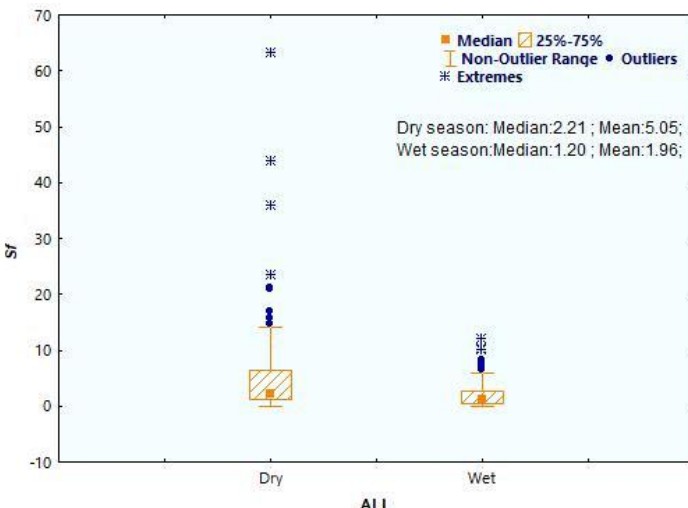

**Figure 2: A comparison of ecohydrological sensitivity in dry season and wet season.**

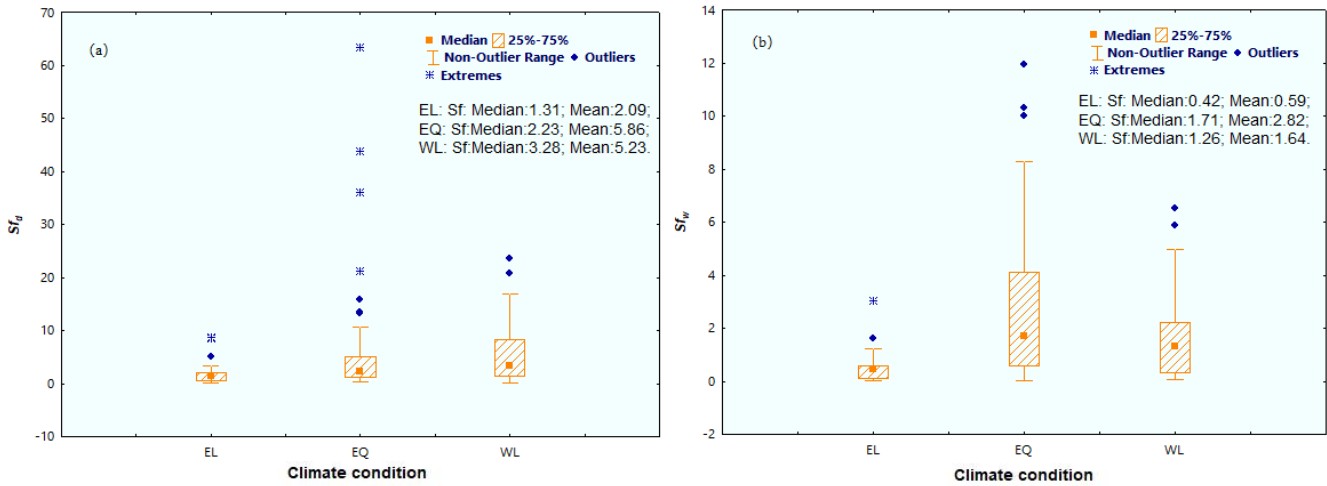


**Figure 3: Comparisons of ecohydrological sensitivity grouped by energy-limited (EL), equitant (EQ) and water-limited (WL) conditions in (a) dry season and (b) wet season. ($S_{fd}$ and $S_{fw}$ are the dry season ecohydrological sensitivity and the wet season ecohydrological sensitivity)**





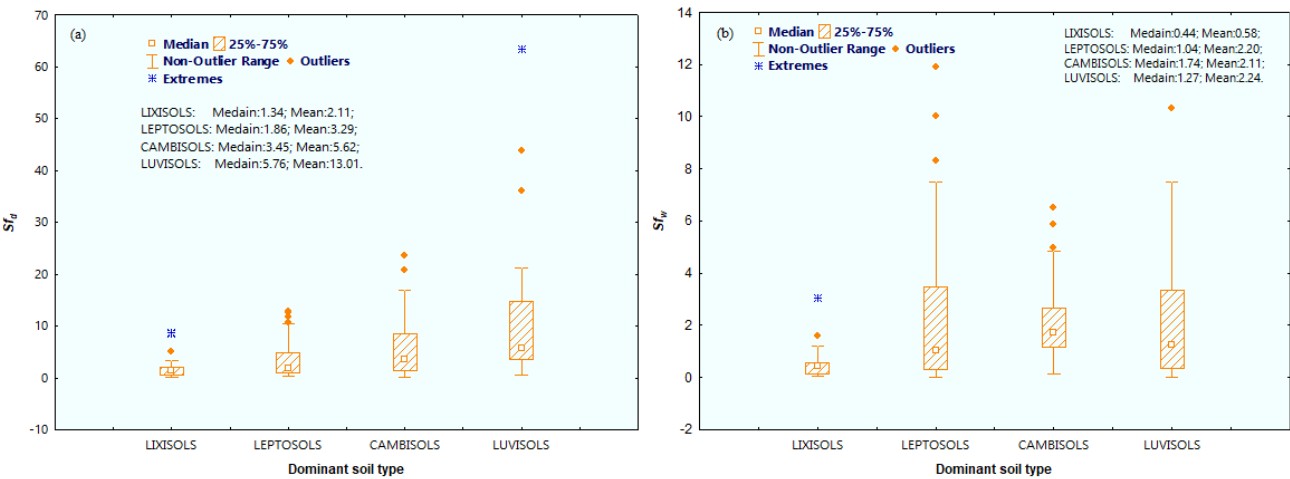


**Figure 4: Comparisons of ecohydrological sensitivity grouped by dominant soil type in (a) dry season and (b) wet season.**

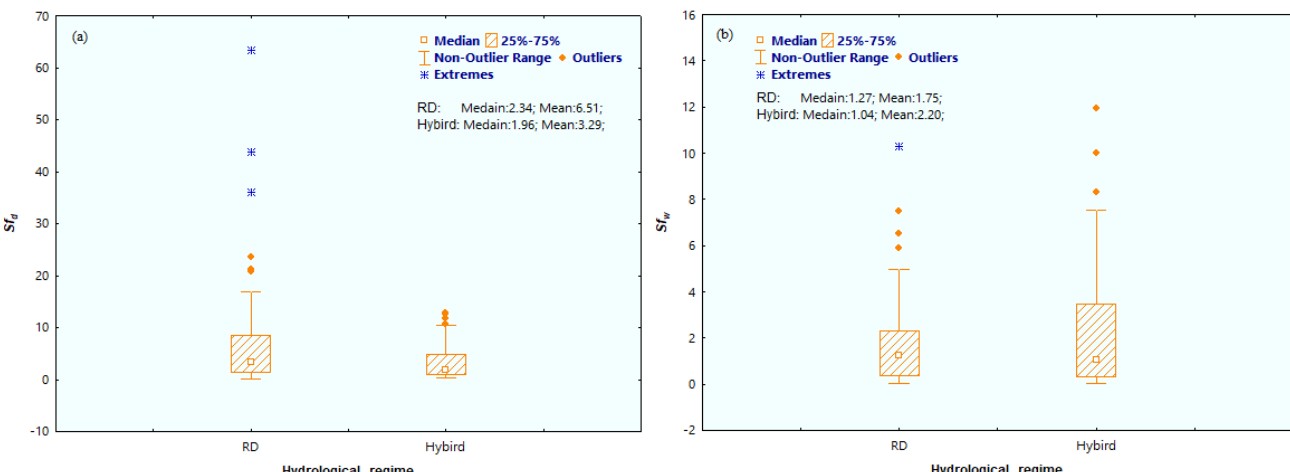

**Figure 5: Comparisons of ecohydrological sensitivity grouped by rain-dominated and hybrid regimes in (a) dry season and (b) wet**
**season.**





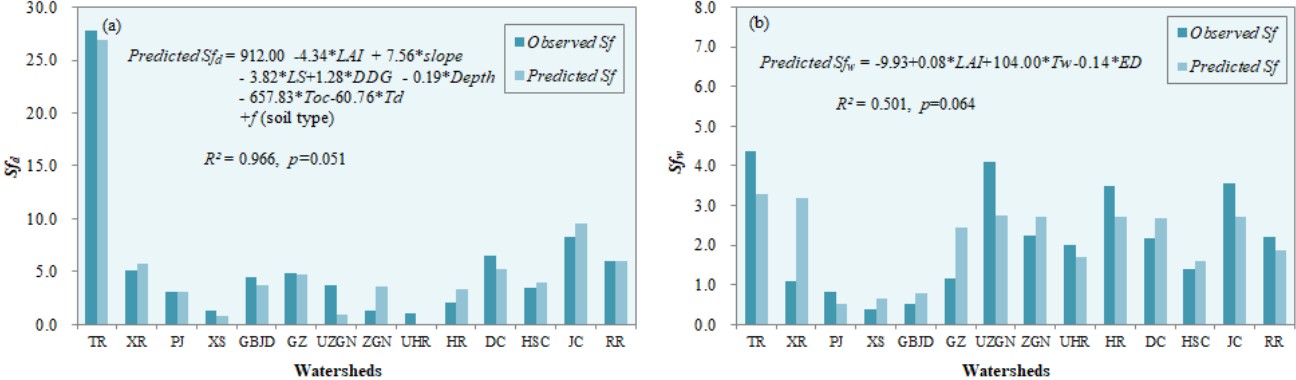

**Figure 6: Comparisons of observed and predicted ecohydrological sensitivity in (a) dry season and (b) wet season. (TR, XR, PJ, XS, GBJD, GZ, UZGN, ZGN, UHR, HR, DC, HSC, JC and RR refer to the Tangwang River, Xinancha River, Pingjiang, Xiangshui, Gongbujiangda, Gengzhang, Upper Zagunao, Zagunao, Upper Heishui River, Heishui River, Dongchuan, Heishuichuan, Jingchuan and Rui River watersheds, respectively)**




**Tables:**

**Table 1: Watershed characteristics in the study watersheds**

| Watersheds | Area (km²) | Mean elevation (m) | Slope (°) | Dry season | | | | | Wet season | | | | |
|---|---|---|---|---|---|---|---|---|---|---|---|---|---|
| | | | | $T_{mean}$ (°C) | P (mm) | ET (mm) | Q (mm) | LAI (m²/m²) | $T_{mean}$ (°C) | P (mm) | ET (mm) | Q (mm) | LAI (m²/m²) |
| Pingjiang | 2778 | 314 | 15.1 | 13.5 | 501.1 | 254.0 | 236.2 | 1.45 | 22.3 | 1310.7 | 585.5 | 604.3 | 1.90 |
| Xiangshui | 1742 | 440 | 17.6 | 14.3 | 472.0 | 274.6 | 242.5 | 2.54 | 22.0 | 1402.7 | 659.5 | 616.4 | 3.17 |
| Tangwang River | 19189 | 447 | 8.7 | -9.6 | 60.3 | 42.4 | 30.1 | 0.73 | 14.8 | 517.1 | 367.4 | 239.9 | 3.53 |
| Xinancha River | 2585 | 507 | 11.3 | -11.1 | 81.0 | 47.5 | 37.3 | 0.71 | 12.8 | 567.5 | 398.9 | 293.6 | 3.59 |
| Upper Zagunao | 2442 | 3814 | 31.0 | 5.3 | 109.3 | 146.2 | 176.9 | 0.83 | 17.0 | 848.6 | 560.4 | 672.9 | 1.87 |
| Zagunao | 4629 | 3622 | 31.7 | 5.0 | 164.4 | 139.9 | 144.0 | 0.86 | 16.6 | 759.9 | 484.2 | 583.2 | 2.14 |
| Upper Heishui River | 1710 | 3858 | 27.8 | -1.8 | 121.1 | 102.7 | 136.9 | 0.50 | 9.7 | 599.8 | 408.6 | 630.0 | 1.78 |
| Heishui River | 7170 | 3619 | 27.3 | -1.8 | 121.1 | 103.0 | 117.8 | 0.53 | 9.7 | 599.8 | 410.5 | 471.2 | 1.93 |
| Gongbujiangda | 6323 | 4946 | 27.2 | 2.5 | 61.0 | 52.3 | 60.8 | 0.11 | 12.9 | 611.8 | 352.4 | 530.7 | 0.45 |
| Gengzhang | 16000 | 4752 | 28.3 | 3.8 | 83.1 | 68.2 | 95.8 | 0.22 | 13.4 | 783.6 | 404.8 | 880.3 | 0.59 |
| Dongchuan | 3049 | 1415 | 16.3 | 0.1 | 68.1 | 59.2 | 6.6 | 0.19 | 16.6 | 438.5 | 321.0 | 23.1 | 0.59 |
| Heshuichuan | 832 | 1340 | 16.8 | 0.9 | 83.5 | 71.1 | 12.4 | 0.34 | 17.0 | 471.9 | 336.8 | 135.1 | 1.51 |
| Jingchuan | 3155 | 1678 | 13.5 | 2.2 | 81.1 | 66.7 | 16.6 | 0.26 | 18.1 | 444.4 | 305.3 | 39.5 | 0.98 |
| Rui River | 1688 | 1608 | 13.0 | 0.1 | 83.6 | 74.0 | 17.6 | 0.28 | 15.2 | 488.3 | 364.0 | 55.1 | 1.23 |

*Note:* $T_{mean}$, P, ET, Q and LAI stand for mean temperature, precipitation, actual evapotranspiration, streamflow and leaf area index during the study period.



**Table 2: Definition or description of the selected influencing factors**

| No. | Category | Abbreviation | Metrics | Definition or description |
|---|---|---|---|---|
| 1 | Climate | *DI* | Dryness index | DI=PET/P, annual potential evaporation (PET) was calculated by Hargreaves method (Hargreaves and Samani, 1985). It shows interactions between energy and water and indicates the water availability for vegetation growth. |
| 2 | | $P_e$ | Effective precipitation | *Pe=P-E*, actual evapotranspiration was calculated by Zhang's equation (Zhang et al., 2001). |
| 3 | Vegetation | LAI | Leaf area index | One-half of the total green leaf area per unit of horizontal ground surface area. Derived from GLASS Product. |
| 4 | | Forest coverage | Forest coverage | Forest coverage in a watershed. |
| 5 | | Vegetation coverage | Vegetation coverage | Vegetation coverage in a watershed (total coverage of forest, shrubland and grassland). |
| 6 | Soil | Soil types | Number of soil types | Total number of soil types in a watershed. |
| 7 | | $T_{oc}$ | Topsoil organic carbon | Amount of carbon bound in human, animal and plant residues and microorganisms formed by microbial action in soil. |
| 8 | | $S_{oc}$ | Subsoil organic carbon | |
| 9 | | $T_{ece}$ | Topsoil salinity | Soil total salinity. |
| 10 | | $S_{ece}$ | Subsoil salinity | |
| 11 | | $T_w$ | Topsoil available water holding capacity | Soil moisture in a stable level. |
| 12 | | $S_w$ | Subsoil available water holding capacity | |
| 13 | | $T_{hy}$ | Topsoil saturated hydraulic conductivity | Infiltration rate of each hydraulic gradient. |
| 14 | | $S_{hy}$ | Subsoil saturated hydraulic conductivity | |
| 15 | | $T_d$ | Topsoil bulk density | Soil mass of each volume. |
| 16 | | $S_d$ | Subsoil bulk density | |
| 17 | Landscape | PN | Patch number | Total number of patches within a specified land cover class. |
| 18 | | PD | Patch density | The number of patches per unit area. |
| 19 | | LPI | Largest patch index | The ratio of the largest patch area to total area. |
| 20 | | ED | Edge density | The total length of patches per unit area. |
| 21 | | CONTAG | Contagion index | Indicates the aggregation of patches. |
| 22 | | SHDI | Shannon's diversity index | Based on information theory, indicates the patch diversity in landscape. |
| 23 | | SIDI | Simpson's diversity index | Indicates the patch diversity in landscape. |




**Table 2: Definition or description of the selected influencing factors (continued)**

| No. | Category | Abbreviation | Metrics | Definition or description |
|---|---|---|---|---|
| 24 | Topographic | Area | Area of a watershed | Area draining to watershed outlet. |
| 25 | | Perimeter | Perimeter of a watershed | Perimeter of a watershed. |
| 26 | | Elevation | Mean elevation | Mean value of all DEM pixels in a watershed. |
| 27 | | ΔElevation | Elevation difference | Difference between the highest elevation and the lowest elevation in a watershed. |
| 28 | | Slope | Average slope | Slope degree of each DEM pixel, can be used in estimation of energy budgets. |
| 29 | | LS | Slope Length Factor | A combined factor of slope length and slope gradient. |
| 30 | | Length | Flow Path Length | The average flow path length starting from the seeds. |
| 31 | | Max Length | Maximum Flow Path Length | The maximum distance of water flow to a point. |
| 32 | | TWI | Topographic Wetness Index | TWI=ln (SCA/tan(slope)), it shows the spatial distribution of zones of surface saturation and soil water content (Ambroise et al., 1996). |
| 33 | | CON | Convergence | Convergence of a cell, which is calculated based on the surrounding eight cells. 100% convergence means all surrounding grid cells flow into the center cell. |
| 34 | | DDG | Downslope distance gradient | An indicator for assessing the impact of the local slope characteristics on a hydraulic gradient. Values are lower on concave slope profiles and higher on convex slope profiles. |
| 35 | | SA | Surface Area | Land area of each DEM. |
| 36 | | TPI | Topographic Position Index | TPI≈0 indicates flat area. TPI>0 tends towards ridge tops and hilltops. TPI< 0 tends towards the valley and canyon bottoms. |
| 37 | | TRI | Terrain Ruggedness Index | The degree of difference in elevation among adjacent cells. |
| 38 | | PO | Topographic Positive Openness | The degree of dominance or enclosure of a location on an irregular surface. Values are high for convex forms. |
| 39 | | NO | Topographic Negative Openness | |
| 40 | | Depth | Valley depth | Difference between the elevation and an interpolated ridge level. |






**Table 3: Classification of watersheds**

| Watersheds | Climate condition | Dominant soil type | Hydrological regime |
|---|---|---|---|
| Pingjiang | Energy-limited | LIXISOLS | Rain-dominated |
| Xiangshui | Energy-limited | LIXISOLS | Rain-dominated |
| Tangwang River | Equitant | LUVISOLS | Rain-dominated |
| Xinancha River | Equitant | LUVISOLS | Rain-dominated |
| Upper Zagunao | Equitant | LEPTOSOLS | Hybrid |
| Zagunao | Equitant | LEPTOSOLS | Hybrid |
| Upper Heishui River | Equitant | LEPTOSOLS | Hybrid |
| Heishui River | Equitant | LEPTOSOLS | Hybrid |
| Gongbujiangda | Water-limited | LEPTOSOLS | Hybrid |
| Gengzhang | Water-limited | LEPTOSOLS | Hybrid |
| Dongchuan | Water-limited | CAMBISOLS | Rain-dominated |
| Heshuichuan | Water-limited | CAMBISOLS | Rain-dominated |
| Jingchuan | Water-limited | CAMBISOLS | Rain-dominated |
| Rui River | Water-limited | CAMBISOLS | Rain-dominated |


**Table 4: Mann Whitney U test for ecohydrological sensitivity between dry season and wet season**

| Season | Z | $p$ |
|---|---|---|
| Dry season vs. Wet season | **5.63** | **0.00*** |

*Note:* The bolded number with **\*** indicates statistically significant at $p < 0.05$.







**Table 5: Mann Whitney U tests for the differences of seasonal ecohydrological sensitivity between climate condition, dominant soil type and hydrological regime**

| Watershed classification | Pairs | $Sf_d$ | | $Sf_w$ | |
|---|---|---|---|---|---|
| | | Z | p | Z | p |
| | EL-EQ | **-2.14** | **0.03*** | **-3.98** | **<0.001*** |
| Climate condition | EL-WL | **-3.09** | **<0.002*** | **-3.15** | **<0.002*** |
| | EQ-WL | -1.41 | 0.16 | **2.20** | **0.03*** |
| | LIXISOLS-LUVISOLS | **-3.70** | **<0.001*** | **-2.19** | **0.028*** |
| | LIXISOLS-LEPTOSOLS | **-1.79** | **0.074*** | **-2.93** | **0.003*** |
| | LIXISOLS-CAMBISOLS | **-2.95** | **0.003*** | **-4.62** | **<0.001*** |
| Dominant soil type | LUVISOLS-LEPTOSOLS | **3.53** | **<0.001*** | 0.02 | 0.98 |
| | LUVISOLS-CAMBISOLS | **1.88** | **0.059*** | -0.80 | 0.42 |
| | LEPTOSOLS-CAMBISOLS | **-2.20** | **0.027*** | -1.42 | 0.15 |
| Hydrological regime | RD-Hybrid | **1.97** | **0.05*** | -0.26 | 0.79 |

*Note:* $Sf_d$ and $Sf_d$ are dry season and wet season ecohydrological sensitivity, respectively; EL, EQ and WL refer to energy-limited, equitant and water-limited watersheds, respectively; RD is Rain-dominated.

The bolded number with * indicates statistically significant at $p < 0.10$.





**Table 6: Correlation analysis between seasonal ecohydrological sensitivities and contributing factors.**

| | Variables | $Sf_d$ Kendall | $a$ | $R^2$ | $Sf_w$ Kendall | $a$ | $R^2$ |
|---|---|---|---|---|---|---|---|
| Climate | DI | **0.44\*** | **2.37\*** | **0.41** | 0.19 | 0.51 | 0.05 |
| | $P_e$ | -0.23 | -0.01 | 0.09 | -0.32 | **-0.03\*** | **0.23** |
| Vegetation | Vegetation coverage | **-0.51\*** | **-0.08\*** | **0.53** | 0.08 | 0.01 | 0.01 |
| | Forest coverage | **-0.36\*** | -0.03 | 0.13 | 0.21 | 0.01 | 0.06 |
| | LAI | **-0.44\*** | **-1.62\*** | **0.05** | 0.09 | -0.08 | 0.00 |
| Topography | Area^c | 0.15 | 0.28 | 0.01 | 0.19 | 0.28 | 0.02 |
| | Perimeter^c | 0.23 | 1.75 | 0.07 | 0.25 | 1.08 | 0.15 |
| | Elevation^c | 0.00 | -0.10 | 0.00 | 0.12 | 0.23 | 0.03 |
| | ΔElevation^c | 0.10 | 0.71 | 0.05 | 0.27 | 0.5 | 0.06 |
| | Slope | **-0.39\*** | **-0.15\*** | **0.28** | -0.03 | -0.01 | 0.00 |
| | LS | **-0.40\*** | **-0.20\*** | **0.24** | 0.04 | 0.48 | 0.01 |
| | Length | -0.18 | $-4.3 \times 10^{-3}$ | 0.10 | 0.21 | $-1.2 \times 10^{-2}$ | 0.03 |
| | Max Length | -0.23 | $-1.9 \times 10^{-3}$ | 0.15 | 0.32 | $-1.4 \times 10^{-3}$ | 0.10 |
| | TWI | **0.62\*** | **4.30\*** | **0.51** | 0.19 | 1.05 | 0.15 |
| | CON | 0.12 | 0.04 | 0.04 | 0.20 | **0.05\*** | **0.20** |
| | DDG | **0.49\*** | **0.10\*** | **0.45** | 0.03 | 0.02 | 0.05 |
| | SA | -0.13 | $1.5 \times 10^{-3}$ | 0.00 | 0.14 | $4.2 \times 10^{-3}$ | 0.15 |
| | TPI | -0.04 | 3.45 | 0.00 | -0.05 | 8.86 | 0.03 |
| | TRI | -0.33 | **-0.32\*** | **0.23** | 0.01 | 0.02 | 0.00 |
| | Positive Openness | **0.36\*** | **14.23\*** | **0.26** | 0.08 | 0.63 | 0.00 |
| | Negative Openness | 0.34 | **14.78\*** | **0.25** | 0.03 | 0.43 | 0.00 |
| | Depth | -0.31 | **-0.01\*** | **0.32** | -0.10 | -0.01 | 0.01 |
| Soil | $T_w$ | 0.25 | 74.26 | 0.05 | **0.53\*** | **125.46\*** | **0.38** |
| | $T_{hy}$ | -0.03 | 0.04 | 0.01 | **0.41\*** | **0.15\*** | **0.25** |
| | $T_d$ | 0.28 | **32.32\*** | **0.28** | 0.10 | 2.49 | 0.01 |
| | $T_{oc}$ | -0.21 | **-3.99\*** | **0.27** | -0.11 | -0.29 | 0.00 |
| | $T_{ece}$ | **0.39\*** | **10.74\*** | **0.28** | 0.30 | 3.99 | 0.19 |
| | $S_w$ | -0.09 | -17.10 | 0.03 | 0.06 | -3.80 | 0.01 |
| | $S_{hy}$ | 0.15 | 0.28 | 0.07 | 0.30 | **0.30\*** | **0.22** |
| | $S_d$ | 0.00 | 13.30 | 0.06 | 0.15 | 8.66 | 0.08 |
| | $S_{oc}$ | 0.17 | 3.80 | 0.03 | -0.09 | 1.76 | 0.02 |
| | $S_{ece}$ | 0.34 | 7.71 | 0.16 | 0.28 | **3.87\*** | **0.22** |
| | Soil types | -0.30 | -0.11 | 0.14 | **0.37\*** | 0.06 | 0.13 |
| Landscape | PN | -0.18 | $-5.2 \times 10^{-4}$ | 0.02 | 0.01 | $-4.2 \times 10^{-5}$ | 0.00 |
| | PD | **-0.54\*** | **-10.83\*** | **0.30** | -0.25 | -4.73 | 0.15 |
| | LPI | 0.08 | 0.02 | 0.04 | 0.06 | 0.04 | 0.00 |
| | ED | **-0.36\*** | -0.27 | 0.17 | -0.32 | **-0.19\*** | **0.23** |
| | CONTAG | 0.03 | 0.02 | 0.01 | 0.10 | 0.03 | 0.09 |
| | SHDI | -0.05 | -0.31 | 0.00 | -0.06 | -0.66 | 0.03 |
| | SIDI | -0.08 | -0.74 | 0.00 | -0.03 | -0.69 | 0.01 |

*Note*: Linear regressions are built as y=ax+b, where *a* is the slope of the linear regression; c means parameters are transferred into ln() format.

The bolded number with * indicates statistically significant at *p* < 0.10.



**710**    **Table 7: Selected factor analysis models**

|  | Influencing factors | MSA | KMO | Bartlett's test |
|---|---|---|---|---|
| Dry season | DI, slope, LS, TWI, DDG, TRI, Depth, NO | ≥0.53 | 0.730 | 0.000 |
| Wet season | $P_e$, CON, $T_w$, $T_{hy}$, $S_{hy}$, ED | ≥0.57 | 0.634 | 0.000 |