# Peer review of "Quantification of Ecohydrological Sensitivities and Their Influencing Factors at the Seasonal Scale"

_Hydrology and Earth System Sciences, 2020_

## Referee Comment (RC1) · Anonymous Referee #1 · 3 Sep 2020

The authors proposed an index called ecohydrological sensitivity, and used many factors to see the impact of catchment characteristics on ecohydrological sensitivity. Honestly, I am not fully convinced to accept such a new term, and its scientific contribution to ecohydrology community.

The method is too superficial, without any new convincing method. Data set is too small, only with 17 basins, it is hard to get solid conclusions. I suggest to involve large number of basins.

The conclusions are either too obvious or too farfetched. For example, the first key finding in dry basins. Sf=deltaQ/(Q*deltaLAI). since Q is small in dry basins. Even

with the same change of deltaQ, the Sf is large anyway. The third one said "3) the dry season ecohydrological sensitivity was mostly determined by topography, soil and vegetation, while the wet season ecohydrological sensitivity was mainly controlled by soil, landscape and vegetation." the only difference between dry and wet season is topography (matters in dry seasons) and landscape (matters in wet seasons). it is hard to accept this conclusion. Does topography or landscape significantly change in dry and wet seasons? With a statistic model, any input data will generate certain relations. But whether the relation has physical meanings or not, which needs more evidences.

In summary, I do not think this work has enough contribution to improve our understanding on ecohydrology, and achieved any significant conclusions which has wide implications.

---

## Author Comment (AC1) · 18 Sep 2020

We greatly appreciate the effort made by Referee #1 in reviewing the paper. Please see our detailed responses to your concerns.

1. The authors proposed an index called ecohydrological sensitivity, and used many factors to see the impact of catchment characteristics on ecohydrological sensitivity. Honestly, I am not fully convinced to accept such a new term, and its scientific contribution to ecohydrology community.

Response: Ecohydrological sensitivity is a useful term for hydrological science and wa-

tershed management. Although the term has often been mentioned in existing studies, there lacks a commonly-accepted definition for its quantitative assessment and comparisons. To our best knowledge, our work is the first study to define and quantify ecohydrological sensitivity at the seasonal scale. Based on our tests and analyses on 14 large watersheds in China, we believe that the ecohydrological sensitivity index developed in this paper provides a useful and common basis for assessing hydrological sensitivity.

It would have been more constructive if referee # 1 could explain why he or she is not fully convinced by this definition. An unfounded statement is not helpful and convincing.

2.The method is too superficial, without any new convincing method. Data set is too small, only with 17 basins, it is hard to get solid conclusions. I suggest to involve large number of basins.

Responses: We strongly disagree with this statement on the method. The method used in this study is a well-established technique for separating the relative contributions of vegetation change and climate variability to seasonal mean flows in any individual watershed. This way, the effects of vegetation change are quantified. This technique developed by our research group has been successfully applied in some Canadian and Chinese watersheds and our relevant works have been published in peer review journals including Water Resources Research, Hydrology and Earth System Sciences, Journal of Hydrology, Ecohydrology (Wei and Zhang, 2010; Zhang et al., 2012; Zhang and Wei, 2012; Liu et al., 2015; Li et al., 2018; Hou et al., 2018; Giles-Hansen et al., 2019). For this study, we applied this technique to 14 watersheds with detailed analyses and results presented in the Supplement. Based on those results from 14 individual watersheds, we assessed hydrological sensitivities and analyzed their contributing factors. I believe that our research approach and the applied method are solid and the conclusions are scientifically supported.

Another important reason why this statement is not convinced is that we used 14 large

watersheds as examples in our study rather than 17. This number is clearly described at the beginning of the abstract, and thus we assume the referee #1 might just quickly go over the manuscript rather than carefully reviewing it. Regarding the dataset from 14 watersheds, we understand the referee # 1's point. We agree that the more watersheds used for the study, the more robust conclusions we can derive. Given tremendous analyses involved for each individual watershed, we need to take a stance between the number of watersheds and the detailed levels of analysis. We think that 14 watersheds should be a reasonable number for this study.

3.The conclusions are either too obvious or too farfetched. For example, the first key finding in dry basins. Sf=deltaQ/(Q*deltaLAI). since Q is small in dry basins. Even with the same change of deltaQ, the Sf is large anyway. The third one said "3) the dry season ecohydrological sensitivity was mostly determined by topography, soil and vegetation, while the wet season ecohydrological sensitivity was mainly controlled by soil, landscape and vegetation." the only difference between dry and wet season is topography (matters in dry seasons) and landscape (matters in wet seasons). it is hard to accept this conclusion. Does topography or landscape significantly change in dry and wet seasons? With a statistic model, any input data will generate certain relations. But whether the relation has physical meanings or not, which needs more evidences.

Response: There is no doubt that the ecohydrological sensitivities depend on climate (Q), but they are also affected by delta Q. According to our analysis, drier regions may not be necessarily more hydrologically sensitive. For example, in our calculations (Table S4 in the Supplement), the dry season Sf in a temperate watershed, the Tangwang River (27.75) is greater than that in the Dongchuan (6.54), Heshuichuan (3.45), Jingchuan (8.27) and Rui River (6.03) watersheds in the Jing River located in semi-arid region, indicating that the hydrological sensitivity to vegetation change in this temperate watershed is greater than that in the watersheds with lower precipitation.

Regarding the comment on the third finding "3) the dry season ecohydrological sensitivity was mostly determined by topography (slope, slope length, valley depth, downslope

distance gradient), soil (topsoil organic carbon, topsoil bulk density) and vegetation (LAI), while the wet season ecohydrological sensitivity was mainly controlled by soil (topsoil available water holding capacity), landscape (edge density) and vegetation (leaf area index)". Response: we think the referee #1 might misunderstand this result. For any individual watersheds, topography is not a driving variable for hydrological changes as it remains unchanged. However, it comes into play when the differences in hydrological response to vegetation change among various watersheds are compared.

References Giles-Hansen, K., Li, Q. and Wei, X.: The Cumulative Effects of Forest Disturbance and Climate Variability on Streamflow in the Deadman River Watershed. Forests 2019, 10(2), 196, https://doi.org/10.3390/f10020196, 2019. Hou, Y., Zhang, M., Meng, Z., Liu, S., Sun, P., and Yang, T.: Assessing the impact of forest change and climate variability on dry season runoff by an improved single watershed approach: A comparative study in two large watersheds, China, Forests, 9, 46, 10.3390/f9010046, 2018. Li, Q., Wei, X., Zhang, M., Giles-Hansen, K., and Wang, Y.: The cumulative effects of forest disturbance and climate variability on streamflow components in a large forest-dominated watershed, J. Hydrol, 557, 448-459. 10.1016/j.jhydrol.2017.12.056, 2018. Liu, W., Wei, X., Liu, S., Liu, Y., Fan, H., Zhang, M., Yin, J. and Zhan, M.: How do climate and forest changes affect long-term streamflow dynamics? A case study in the upper reach of Poyang River basin, Ecohydrol., 8, 46-57, doi: 10.1002/eco.1486, 2015. Wei, X., and Zhang, M.: Quantifying streamflow change caused by forest disturbance at a large spatial scale: A single watershed study, Water Resour. Res., 46, 10.1029/2010wr009250, 2010. Zhang, M., Wei, X., Sun, P., and Liu, S.: The effect of forest harvesting and climatic variability on runoff in a large watershed: The case study in the Upper Minjiang River of Yangtze River basin, J. Hydrol., 464, 1-11, 10.1016/j.jhydrol.2012.05.050, 2012. Zhang, M., and Wei, X.: The effects of cumulative forest disturbance on streamflow in a large watershed in the central interior of British Columbia, Canada, Hydrol. Earth Syst. Sci., 16, 2021-2034, 10.5194/hess-16-2021-2012, 2012. Zhang, M., Liu, N., Harper, R., Li, Q., Liu, K., Wei, X.., Ning, D., Hou, Y., and Liu, S.: A global review on hydrological responses to forest change across multiple spatial scales: Importance of scale, climate, forest type and hydrological regime, J. Hydrol., 546, 44-59, 10.1016/j.jhydrol.2016.12.040, 2017.

---

## Referee Comment (RC2) · Anonymous Referee #2 · 3 Oct 2020

Sorry for the late review on the manuscript by Hou et al.. It is a very interesting study to define the eco-hydrological sensitivity and explore its influencing factors at seasonal scale, using 14 watersheds across climate, soil, topography and landscape gradients in Mainland China. The unique perspective of eco-hydrological sensitivity is attractive and promising, and the results are potentially significant. The article is well written with clear logic and detailed analysis. Overall, I think this article is worthy of publishing. However, there are a couple of major concerns before I recommend for publishing the research. First of all, the methods and data usage are not very much clearly presented. For instance, the definition of dry and wet season is of critical importance as the study covers the climate regions from subtropical in southern China and cold temperate region in northeast China. However, authors do not clearly present. Similarly, it seems not clear how the dry and wet season LAIs were calculated for the watersheds located in the very different climate regions? Second, the justification for using large number of topographic and landscape indexes is missing. In reality, almost every feature in the watershed will have impact on the watershed responses, even though some can be ruled out and some are relevant than others mathematically. Thirdly, 14 watersheds studied are subjected to different disturbance regimes hydrologically and ecologically, yet, the separation of stream change in dry and wet season into vegetation change and climate change seems not very much convincing. This point also should be addressed or at least the weakness of current study should be indicated in the discussion and/or conclusion sections.

Specific points are as follows: Line 43: Please make a clear distinction between the dry and the wet season in your study. Line 49: 'shown' is more common. Line 65: Please delete 'in spite of its usefulness'. Line 89-92: Here, the logic is elusive. The background is right in China. But this doesn't mean a good opportunity to explore the index on a short time scale (e.g., seasonal scale). Line 98-99: What were the selection criteria for fourteen large watersheds? And their representativeness is not distinct. Line 115: How can the author define the equally divided dry and wet seasons for the watersheds located in the very much different climate zones across China? This is critical, please specify clearly. Line 117-118: Please give the PET formula. Line 121-123: How to reclassify land cover types? What is the basis for this? Line 124-134: Have you compared the two RS products with observations in the fieldïij§Which is more accurate for your studyïij§ Line 149-150: Remove it. Line 154: Repeat. Remove 'which is calculated by the improved single watershed approach'. Line 157: How do you consider the auto-correlation between the influencing drivers? Line 189: Why estimate the significant at a level of 0.10 rather than usual 0.01 or 0.05? Line 199: How do you consider the interaction effects? Or if there is collinearity between the selected variables? How to overcome this problem? Figures 2-5: They cannot display the differences intuitively and clearly. Please redraw these figures. Line 331: should be

'a dominating factor'. Line 378-379: What are the uncertainties of the simple multiple linear model for providing a reliable and robust assessment framework based on the selected fourteen watersheds in China? I do believe that authors should address this issue.

---

## Author Comment (AC2) · 28 Oct 2020

We highly appreciate your constructive comments and suggestions which are of great help to us in revising our manuscript. We have carefully addressed all your comments. Please find our detailed responses below.

Responses to major comments:

Comment 1: The methods and data usage are not very much clearly presented. For instance, the definition of dry and wet season is of critical importance as the study covers the climate regions from subtropical in southern China and cold temperate region

in northeast China. However, authors do not clearly present. Similarly, it seems not clear how the dry and wet season LAIs were calculated for the watersheds located in the very different climate regions?

Response: Our submission includes the main text and supplement. The definition and distinction of the dry season and wet season for each watershed are provided in the Supplement S1 which also includes the data descriptions on climate, topography, soil, land cover, and LAI in the selected watersheds. Meanwhile, the method for quantifying the effect of vegetation change on seasonal streamflow is described in detail in Supplement S2, which is the basis for the quantification of seasonal ecohydrological sensitivity. Regarding the calculation of watershed LAI, we have generated two data series of LAI: dry season LAI (mean value of the LAIs in the dry season) and wet season LAI (mean value of the LAIs in the wet season) from the entire study period. To address your concerns, we will modify the sections of study watershed, data and method to present our methodology more clearly by e.g., adding a brief description on the definition and distinction of the dry season and wet season for each watershed and the calculation in our revised manuscript.

Comment 2: The justification for using large number of topographic and landscape indexes is missing. In reality, almost every feature in the watershed will have impact on the watershed responses, even though some can be ruled out and some are relevant than others mathematically.

Response: In this study, seventeen topographic indices and seven landscape indices are involved to represent topographic and landscape conditions in watersheds. Firstly, these indices have been identified based on previously published studies, which are most frequently used in studying the topographic effect on hydrological processes. We agree with you that every feature in a watershed will have a certain impact on the watershed responses. For example, area, perimeter, mean elevation, and elevation differences provide basic topographic conditions for each watershed, showing watershed heterogeneity. Slope, flow path length (Length), and slope length factor (LS) are

indices used for assessing erosion hazard. Topographic wetness index (TWI) is a critical topographic index related to soil water content and surface saturation. Shannon's diversity index (SHDI) and Simpson's diversity index (SIDI) could be applied to indicate a patch diversity of landscape. As you mentioned, some of them can be ruled out while some of them are more relevant than others to watershed responses. However, too many predicting factors are likely to increase the redundancy of a prediction model. A model with more predicting factors does not guarantee a more accurate prediction. For example, some of these indices are highly linearly related to others, which will lead to a multicollinearity problem in a prediction model. Thus, multicollinear relationships between these indices must be detected and confirmed first and then to identify the key factors that are mostly related hydrological response to vegetation change by factor analysis and step-wise regression. The whole selection process is a trade-off between the model complexity and model performance, which may bring some uncertainties. We will add the justification for using large number of topographic and landscape indices and discuss the associated uncertainties in our revised manuscript.

Comment 3: 14 watersheds studied are subjected to different disturbance regimes hydrologically and ecologically, yet, the separation of stream change in dry and wet season into vegetation change and climate change seems not very much convincing. This point also should be addressed or at least the weakness of current study should be indicated in the discussion and/or conclusion sections.

Response: We just briefly describe the method for separating the effects of vegetation change and climate variability on season streamflow in the main text. A more detailed description on the methodology is provided in the Supplement S2. We will modify the method section to describe the methodology more clearly. As you pointed out, 14 large watersheds have been experienced different disturbances, such as vegetation removal, vegetation restoration and anthropogenic activities. It is very challenging to differentiate the hydrological impact of vegetation change, climate change and other watershed disturbances. In this study, seasonal streamflow variations are attributed

to climate variability, vegetation change and other factors. The modified double mass curve (MDMC) is firstly used to remove the effects of climate variability on seasonal streamflow variation. The multivariate ARIMA (ARIMAX) model is then used to quantify seasonal streamflow variation attributed to non-climatic factors (vegetation change and other factors). The 95% confidence intervals (95%CIs) criterion is applied to separate the statistical errors and the seasonal streamflow variation attributed to other factors. The seasonal streamflow variation caused by vegetation change can be quantified eventually. We believe this framework is a feasible methodology for identifying the effect of vegetation change, climate variability and other factors. However, there is no perfect methodology. First, an important assumption of this method is that the vegetation-water relationship during the study period should be stationary, which may be invalid if vegetation-water relationship is nonstationary. In addition, various watershed disturbances such as urbanization, dam regulations and other human activities are considered as a whole (other factors). Therefore, the impact of each watershed disturbance (e.g., urbanization, dam regulation, and irrigation) cannot be quantified separately. We will discuss these limitations of our methodology in the revised manuscript as your suggestion.

Responses to specific comments:

Line 43: Please make a clear distinction between the dry and the wet season in your study.

Response: We will provide a brief description on the distinction between the dry and wet season in the main text as your suggestion.

Line 49: 'shown' is more common.

Response: We will revise it as your suggestion.

Line 65: Please delete 'in spite of its usefulness'.

Response: We will revise it as your suggestion.

Line 89-92: Here, the logic is elusive. The background is right in China. But this doesn't mean a good opportunity to explore the index on a short time scale (e.g., seasonal scale).

Response: We will modify the statement to provide a more convincing justification for assessing ecohydrological sensitivity in watersheds across different climatic zones in China. China has experienced substantial and dynamic vegetation change over the past few decades. Deforestation and biomass loss dominated vegetation change from 1950s to 1980s (Wei et al., 2008), while several nation-wide revegetation programs have been implemented since 1980s (Li et al., 2018b). These large-scale vegetation changes will inevitably impact local and regional water cycles. However, given the large variations in climate, vegetation, soil, topography and landscapes in China, hydrological responses to vegetation change can be highly variable among watersheds. As is known, it is very challenging and time-consuming to assess the hydrological impact of vegetation change in every watershed. There is a need to develop a general framework for an efficient evaluation of the hydrological sensitivity to vegetation change at a watershed scale, which will benefit future water and forest resources management.

Line 98-99: What were the selection criteria for fourteen large watersheds? And their representativeness is not distinct.

Response: We will clarify our selection criteria for the study watersheds in the revised manuscript. Given the great difficulty that there is no free access for hydrological data in China, the number of study watersheds cannot be as large as we want. That means the best strategy for us is to locate a number of representative watersheds based on their hydrological data availability, watershed size, climate type and vegetation type. Given that the dominant climate zones in China include subtropical monsoon, alpine, temperate monsoon and temperate monsoon climate zones, there will be several representative study watersheds in each climate zone. The selected watersheds in each climatic zone are with watershed size greater than 500 km2 along with long-time hydrological data available to meet the data requirements for statistical analysis ($\geq$15

yrs). In addition, we only focus on vegetative watersheds with vegetation cover greater than 30% since less vegetated watersheds are mostly located in arid regions where the effect of vegetation change on streamflow is not dominant at a watershed scale. With these criteria, we select at least two large watersheds in each climate zone. We will also clarify these criteria in Section 2.1 in the main text. In addition, the detailed descriptions of the study watersheds about their representativeness on climate, topography, soil, dominated vegetation type and hydrological regime are described in the Supplement S1.

Line 115: How can the author define the equally divided dry and wet seasons for the watersheds located in the very much different climate zones across China? This is critical, please specify clearly.

Response: In this study, the dry and wet seasons are defined according to the long-term mean monthly precipitation within a hydrological year (November to October). For watersheds in subtropical monsoon climate (Pingjiang and Xiangshui), a wet season with more than 70% of annual precipitation falling in watersheds covers a period from March to August, while the dry season starts from September to February. In other watersheds from the alpine, temperate monsoon, and temperate continental climate zones, the wet season (also called rainy season mostly in summer) starts from May to August, while dry season is from November to April. We will specify how dry and wet seasons are divided in the revised manuscript as your suggestion.

Line 117-118: Please give the PET formula.

Response: We will provide it as your suggestion.

Line 121-123: How to reclassify land cover types? What is the basis for this?

Response: There are 17 types of land covers in MODIS MCD12Q, including evergreen needleleaf forests, deciduous needleleaf forests, evergreen broadleaf forests, deciduous broadleaf forests, mixed forests, closed shrublands, opened shrublands, woody

savannas, savannas, grasslands, permanent wetlands, croplands, urban and built-up land, cropland/natural vegetation mosaics, permanent snow and ice, barren, and water bodies. We reclassify them into forest (evergreen needleleaf forests, deciduous needleleaf forests, evergreen broadleaf forests, deciduous broadleaf forests and mixed forests), shrubland (closed shrublands and opened shrublands), grassland (woody savannas, savannas and grasslands), agricultural (croplands and cropland/natural vegetation mosaics), snow (permanent snow and ice) and other lands (permanent wetlands, urban and built-up land, barren, and water bodies). Vegetation coverage including forest, shrubland and grassland can be then calculated. We will state how we reclassify the land cover types in the revised manuscript.

Line 124- 134: Have you compared the two RS products with observations in the field? Which is more accurate for your study?

Response: We understand that it will be helpful if the accuracy of the two RS products can be validated by field observations from our study watersheds. However, it is very challenging and expensive to conduct field observations in our study watersheds that distribute across large climatic gradients. Actually, these two RS products have been widely used and have already been validated by some studies in China (Xiao et al., 2016; Yang et al., 2017). Therefore, according to previous studies, we believe the application of the two RS products can be acceptable and reliable. We will discuss the uncertainty associated with the RS products in the revised manuscript. In fact, the two RS products (GLASS LAI and MODIS MCD12Q1) are used for different purposes in our study. The GLASS LAI is used for quantifying watershed vegetation level that is then used to calculate ecohydrological sensitivity, while the MODIS MCD12Q1 is applied for calculating forest and vegetation coverage that is then used to calculate landscape indices by FRAGSTATS 4.2. Therefore, we cannot conclude which product is more accurate for our study.

References Xiao, Z., Liang, S., Wang, J., Xiang, Y., Zhao, X., and Song, J.: Long-Time-Series Global Land Surface Satellite Leaf Area Index Product Derived From MODIS

and AVHRR Surface Reflectance, IEEE T. Geosci. Remote Sens., 54 (9), 5301-5318, 10.1109/TGRS.2016.2560522, 2016. Yang, Y., Xiao, P., Feng, X., and Li, H.: Accuracy assessment of seven global land cover datasets over China, ISPRS Journal of Photogramm. 125, 156-173, https://doi.org/10.1016/j.isprsjprs.2017.01.016, 2017

Line 149-150: Remove it.

Response: We will revise it as your suggestion.

Line 154: Repeat. Remove 'which is calculated by the improved single watershed approach'.

Response: We will revise it as your suggestion.

Line 157: How do you consider the auto-correlation between the influencing drivers?

Response: If we understand right, you mean the influencing drivers are correlated with each other. We have selected 40 indices that are classified into five types including climate, vegetation, topography, soil and landscape in this study. There is no doubt that some of them are correlated with each other. To address this issue, we have firstly performed Kendall correlation analysis and linear regression to identify statistically significant correlations between seasonal ecohydrological sensitivities and 40 indices, where the insignificant indices were excluded for the prediction model. Then we have performed the factor analysis to further reduce the redundancy of indices. Factor analysis can reduce a large number of variables into fewer numbers of factors with important information being retained, which is similar to principal component analysis. Eventually, only a few indices with key influences on seasonal ecohydrological sensitivity are retained for multiple linear regression. In this way, the correlation between the influencing drivers could be greatly reduced. We will provide a clearer description on variables selection to clarify this issue in the revised manuscript.

Line 189: Why estimate the significant at a level of 0.10 rather than usual 0.01 or 0.05?

Response: The significance level for a given hypothesis test is a value for which a pvalue less than or equal to the significance is considered statistically significant. Typical values for the significance level are 0.10, 0.05, and 0.01, indicating that strong evidence against the null hypothesis, and the null hypothesis is rejected accordingly. Meanwhile, there are less than 10%, 5% and 1% probability that the null hypothesis is true. In other words, there are 10%, 5% and 1% probability that a wrong rejection of the null hypothesis could happen, respectively. We understand a lots of studies choose the a significance level of 0.01 and 0.05, while there are also many studies use 0.1 as a significance level according to their research purposes and sample size. In this study, given our sample size is relatively small, we choose a significance level of 0.1 in the correlation analysis in order to identify potential factors with significant influences on seasonal ecohydrological sensitivity as many as possible. Since it is used in the initial selection of influencing factors, we believe a significance level of 0.1 is acceptable.

Line 199: How do you consider the interaction effects? Or if there is collinearity between the selected variables? How to overcome this problem?

Response: Yes, collinearity should be considered in our analysis. As we mentioned before, we have applied correlation analysis and factor analysis to identify key factors as the input for multiple linear regression. Thus, only a few indices with key influences on seasonal ecohydrological sensitivity are retained for multiple linear regression. In this way, the correlation between the influencing drivers could be greatly reduced, which help us to establish a model without collinearity more easily. In addition, we have tested the collinearity of inputting variables for the multiple linear regression. Since the VIF (Variance Inflation Factor) is less than 10, we believe there is no collinearity between the inputting variables for the prediction model. We will clarify this as your suggestion.

Figures 2-5: They cannot display the differences intuitively and clearly. Please redraw these figures.

Response: We will revise these figures as your suggestion.

Line 331: should be 'a dominating factor'.

Response: We will revise it as your suggestion.

Line 378-379: What are the uncertainties of the simple multiple linear model for providing a reliable and robust assessment framework based on the selected fourteen watersheds in China? I do believe that authors should address this issue.

Response: Yes, we totally agree with you that there are some uncertainties and limitations associated with the model. We will discuss these uncertainties as your suggestion. The first limitation could be the sample size. Our models are generated from only 14 large watersheds. We understand more watersheds used for the study will lead to more robust conclusions. However, as we mentioned before, there is a great difficulty in free access to hydrological data in China, it is impossible for us to get an ideal sample size. Besides, the quantification of vegetation impact on seasonal streamflow involves tremendous analyses for each watershed, and there is a trade-off between the number of watersheds and workload. The second limitation could be that our models fail to capture some non-linear relationships between ecohydrological sensitivity and its influencing factors. Other methodologies such as machine learning or neural network could be applied to explore non-linear relationships between ecohydrological sensitivity and its influencing factors with a bigger sample size in future studies.
* * *

---

## Author Response (AR1)

**Response to comments from editor and reviewers**

Dear editor and reviewers,

We sincerely appreciate your constructive suggestions and comments for our manuscript (ID: HESS-2020-336) entitled "*Quantification of Ecohydrological Sensitivities and Their Influencing Factors at the Seasonal Scale*". We have carefully studied them and addressed accordingly. We believe that the revisions that have been made to the manuscript, following the editor and reviewers' comments and suggestions, can lead to improved interpretations of our findings. Our responses and corresponding edits to the manuscript are provided below following the editor and reviewers' original comments highlighted in blue. Please let us know if there is anything else we can do to help further the review process.

Thank you very much for your consideration,

Mingfang Zhang, Ph.D.

Professor, Head of Department of Environmental Science and Engineering

School of Resources and Environment

University of Electronic Science and Technology of China

\*\*\*\*\*\*\*\*\*\*\*\*\*\*\*\*\*\*\*\*\*\*\*\*\*\*\*\*\*\*\*\*\*\*\*\*\*\*\*\*\*\*\*\*\*\*\*\*\*\*\*\*\*\*\*\*\*\*\*\*\*\*\*\*\*\*\*\*\*\*\*\*\*\*\*\*\*\*\*\*\*\*\*\*\*\*\*

**Editor**

Comments:

(1) The author response to Reviewer #1, Comment #1 contains some nice statements about the novelty and contribution of the work that I could not find in the discussion paper. I would encourage you to add sentences #2-3 of this response to the end of line 88 in the discussion paper.

**Response**: Thanks. We mentioned the novelty and contribution in lines 65-66 (lines 71-72 in the revised version) of the discussion paper. As your suggestion, we deleted these sentences of line 65-66 (lines 71-72 in the revised version) and added sentences #2-3 of this response to the end of line 95 in the discussion paper to strengthen the statement about novelty and contribution of our work in the 'Introduction' section. Please see lines 95-98 in the revised version. We also mentioned the novelty and innovation of our study in Abstract (Lines 25-26).

(2) Reviewer #1 raises an interesting point in their Comment #3, which is also similarly raised by Reviewer #2 Comment #3 regarding the reasons for the separation of dry and wet seasons. I think the author response to Reviewer #1 is very interesting and makes an important point that the delta Q is more critical than the Q value itself in the sensitivity. I would ask the authors to consider emphasizing this point in the revision, as I believe it helps to address the comments of both Reviewers on this point.

**Response**: Thanks for this suggestion. We added some descriptions on season separation in lines 124-128. We also emphasized the importance of delta Q in ecohydrological sensitivity in the 'Method' section (sect. 3.1, lines 197-202).

\*\*\*\*\*\*\*\*\*\*\*\*\*\*\*\*\*\*\*\*\*\*\*\*\*\*\*\*\*\*\*\*\*\*\*\*\*\*\*\*\*\*\*\*\*\*\*\*\*\*\*\*\*\*\*\*\*\*\*\*\*\*\*\*\*\*\*\*\*\*\*\*\*\*\*\*\*\*\*\*\*\*\*\*\*\*\*

**Reviewer 1**

Comments:

(1) The authors proposed an index called ecohydrological sensitivity, and used many factors to see the impact of catchment characteristics on ecohydrological sensitivity. Honestly, I am not fully convinced to accept such a new term, and its scientific contribution to ecohydrology community.

**Response**: Ecohydrological sensitivity is a useful term for hydrological science and watershed management. Although the term has often been mentioned in existing literatures, there is general lack of a commonly-accepted definition for its quantitative assessment and comparisons. To our best knowledge, the work is the first study to define and quantify ecohydrological sensitivity at the seasonal scale. Based on our tests and analyses from 14 watersheds in China, we believe that the

ecohydrological sensitivity index developed in this paper provides a useful and common basis for assessing hydrological sensitivity. We primarily described the importance and current understanding of ecohydrological sensitivity in the 'Introduction' section from line 61 to line 83 and emphasized them in lines 95-98 in the revised version.

(2) The method is too superficial, without any new convincing method. Data set is too small, only with 17 basins, it is hard to get solid conclusions. I suggest to involve large number of basins.

**Responses**: We strongly disagree with this statement on the method. The method used in this study is a well-established technique for separating the relative contributions of forest change and climate variability to annual mean flows in any individual watersheds. This way, the effects of forest disturbance or change are quantified. The technique is developed by this group and has been successfully applied in some Canadian and Chinese watersheds (Wei and Zhang, 2010; Zhang et al., 2012; Zhang and Wei, 2012; Liu et al., 2015; Li et al., 2018; Giles-Hansen et al., 2019; Hou et al., 2018). For this study, we applied this technique to 14 watersheds with detailed analyses and results presented in the Supplement (Section S2). Based on those results from 14 individual watersheds, we compared hydrological sensitivities and analyzed their contributing factors. We believe that our research approach and the applied method are sound, and the conclusions are scientifically supported. In the revised version, we briefly described the quantification method in lines 183-190.

Regarding the dataset from 14 watersheds, we understand your point. We agree that the more watersheds used for the study, the more robust conclusions we can derive. Given tremendous analyses involved for each individual watershed, we need to take a stance between the number of watersheds and the detailed levels of analysis. We think that 14 watersheds should be a reasonable number for this study. In the revised manuscript, we discussed the limitations and uncertainties about the sample size. Please see lines 468-471 (Sect. 5.7).

Response: There is no doubt that the ecohydrological sensitivities depend on climate (Q), but they are also affected by delta Q. We emphasized this in lines 197-202 in the revised version. However, drier regions may not be necessarily more sensitive according to our analysis. For example, in our calculations (Table S4 in the Supplement), the dry season *Sf* in a temperate watershed, the Tangwang River (27.75) is greater than those in the Dongchuan (6.54), Heshuichuan (3.45), Jingchuan (8.27) and Rui River (6.03) watersheds in the Jing River located in a semi-arid region, suggesting that the ecohydrological sensitivity to vegetation change in this temperate watershed is greater than those in the watersheds with lower precipitation.

Regarding the comment on the third finding "*3) the dry season ecohydrological sensitivity was mostly determined by topography (slope, slope length, valley depth, downslope distance gradient), soil (topsoil organic carbon, topsoil bulk density) and vegetation (LAI), while the wet season ecohydrological sensitivity was mainly controlled by soil (topsoil available water holding capacity), landscape (edge density) and vegetation (leaf area index)*", we think you might misunderstand this

result. For any individual watersheds, topography is not a driving variable for hydrological changes as it remains unchanged. However, it comes into play when the differences among individual watersheds are compared.

\*\*\*\*\*\*\*\*\*\*\*\*\*\*\*\*\*\*\*\*\*\*\*\*\*\*\*\*\*\*\*\*\*\*\*\*\*\*\*\*\*\*\*\*\*\*\*\*\*\*\*\*\*\*\*\*\*\*\*\*\*\*\*\*\*\*\*\*\*\*\*\*\*\*\*\*\*\*\*\*\*\*\*\*\*\*\*

**Reviewer 2**

Major comments:

(1) The methods and data usage are not very much clearly presented. For instance, the definition of dry and wet season is of critical importance as the study covers the climate regions from subtropical in southern China and cold temperate region in northeast China. However, authors do not clearly present. Similarly, it seems not clear how the dry and wet season LAIs were calculated for the watersheds located in the very different climate regions?

**Response:** Our submission includes the main text and supplement. The definition and distinction of the dry season and wet season for each watershed are provided in the Supplement S1 which also includes the data descriptions on climate, topography, soil, land cover, and LAI in the selected watersheds. We added a brief description on the definition and distinction of the dry season and wet season for each watershed and the calculation in our revised manuscript. Please see lines 124-128 and Table 1 on page 34. Meanwhile, we also described how to quantify seasonal hydrological responses to vegetation change in lines 183-190 in the revised version to give a basic understanding of the quantification approach. This method is described in detail in the Supplement sect. S2, which is the basis for the quantification of seasonal ecohydrological sensitivity. Regarding the calculation of watershed LAI, we have generated two data series of LAI: dry season LAI (mean value of the LAIs in the dry season) and wet season LAI (mean value of the LAIs in the wet season) from the entire study period. Please see lines 171-173.

(2) The justification for using large number of topographic and landscape indexes is missing. In reality, almost every feature in the watershed will have impact on the watershed responses, even though some can be ruled out and some are relevant than others mathematically.

**Response:** In this study, seventeen topographic indices and seven landscape indices are involved to represent topographic and landscape conditions in watersheds. Firstly, these indices have been identified based on previously published studies, which are most frequently used in studying the topographic effect on hydrological processes. We agree with you that every feature in a watershed will have a certain impact on the watershed responses. For example, area, perimeter, mean elevation, and elevation differences provide basic topographic conditions for each watershed, showing

watershed heterogeneity. Slope, flow path length (Length), and slope length factor (LS) are indices used for assessing erosion hazard. Topographic wetness index (TWI) is a critical topographic index related to soil water content and surface saturation. Shannon's diversity index (SHDI) and Simpson's diversity index (SIDI) could be applied to indicate a patch diversity of landscape. As you mentioned, some of them can be ruled out while some of them are more relevant than others to watershed responses. However, too many predicting factors are likely to increase the redundancy of a prediction model. A model with more predicting factors does not guarantee a more accurate prediction. For example, some of these indices are highly linearly related to others, which will lead to a multicollinearity problem in a prediction model. Thus, multicollinear relationships between these indices must be detected and confirmed first and then to identify the key factors that are mostly related hydrological response to vegetation change by factor analysis and stepwise regression. The whole selection process is a trade-off between the model complexity and model performance, which may bring some uncertainties. We added the justification for using large number of topographic and landscape indices in lines 231-245 and discuss the associated uncertainties in our revised manuscript in lines 472-486.

(3) 14 watersheds studied are subjected to different disturbance regimes hydrologically and ecologically, yet, the separation of stream change in dry and wet season into vegetation change and climate change seems not very much convincing. This point also should be addressed or at least the weakness of current study should be indicated in the discussion and/or conclusion sections.

**Response:** We added brief descriptions on the method for separating the effects of vegetation change and climate variability on season streamflow in the main text (lines 183-190). A more detailed description on the methodology is provided in the Supplement S2. As you pointed out, 14 large watersheds have been experienced different disturbances, such as vegetation removal, vegetation restoration and anthropogenic activities. It is very challenging to differentiate the hydrological impact of vegetation change, climate change and other watershed disturbances. In this study, seasonal streamflow variations are attributed to climate variability, vegetation change and other factors. The modified double mass curve (MDMC) is firstly used to remove the effects of climate variability on seasonal streamflow variation. The multivariate ARIMA (ARIMAX) model is then used to quantify seasonal streamflow variation attributed to non-climatic factors (vegetation change and other factors). The 95% confidence intervals (95%CIs) criterion is applied to separate the statistical errors and the seasonal streamflow variation attributed to other factors. The seasonal streamflow variations caused by vegetation change can be quantified eventually. We believe this framework is a feasible methodology for identifying the effect of vegetation change, climate variability and other factors.

However, there is no perfect methodology. Firstly, an important assumption of this method is that the vegetation-water relationship during the study period should be stationary, which may be invalid if vegetation-water relationship is nonstationary. In addition, various watershed disturbances such as urbanization, dam regulations and other human activities are considered as a whole (other factors). Therefore, the impact of each watershed disturbance (e.g., urbanization, dam regulation, and irrigation) cannot be quantified separately. We discussed these limitations of our methodology in the revised manuscript according to your suggestion in lines 460-467.

Specific comments:

(1) Line 43: Please make a clear distinction between the dry and the wet season in your study.

**Response**: Done. We defined dry season and wet season in these 14 watersheds in lines 124-128 and Table 1 on page 34.

(2) Line 49: 'shown' is more common.

**Response**: Done. Thanks!

(3) Line 65: Please delete 'in spite of its usefulness'.

**Response**: We deleted it.

(4) Line 89-92: Here, the logic is elusive. The background is right in China. But this doesn't mean a good opportunity to explore the index on a short time scale (e.g., seasonal scale).

**Response**: We modified the statement to provide a more convincing justification for assessing ecohydrological sensitivity in watersheds across different climatic zones in China. Please see lines 101-106. China has experienced substantial and dynamic vegetation change over the past few decades. Deforestation and biomass loss dominated vegetation change from the 1950s to 1980s (Wei et al., 2008), while several nation-wide revegetation programs have been implemented since the 1980s (Li et al., 2018b). These large-scale vegetation changes will inevitably affect local and regional water cycles. However, given the large variations in climate, vegetation, soil, topography and landscapes in China, hydrological responses to vegetation change can be highly variable among watersheds. As is known, it is very challenging and time-consuming to assess the hydrological impact of vegetation change in every watershed. There is a need to develop a general framework for an efficient evaluation of the hydrological sensitivity to vegetation change at a watershed scale, which will benefit future water and forest resources management.

(5) Line 98-99: What were the selection criteria for fourteen large watersheds? And their representativeness is not distinct.

**Response**: We clarified our selection criteria for the study watersheds in the revised manuscript (lines 113-119 in sect. 2.1). Given the great difficulty that there is no free access for hydrological data in China, the number of study watersheds cannot be as large as we want. That means the best strategy for us is to locate a number of representative watersheds based on their hydrological data availability, watershed size, climate type and vegetation type. Given that the dominant climate zones in China include subtropical monsoon, alpine, temperate monsoon and temperate monsoon climate zones, there will be several representative study watersheds in each climate zone. The selected watersheds in each climatic zone are with watershed size greater than 500 km$^2$ along with long-time hydrological data available to meet the data requirements for statistical analysis ($\geq$15 yrs). In addition, we only focus on vegetative watersheds with vegetation cover greater than 30% since less vegetated watersheds are mostly located in arid regions where the effect of vegetation change on streamflow is not dominant at a watershed scale. With these criteria, we select at least two large watersheds in each climate zone. In addition, the detailed descriptions of the study watersheds about their representativeness on climate, topography, soil, dominated vegetation type and hydrological regime are described in the Supplement S1.

(6) Line 115: How can the author define the equally divided dry and wet seasons for the watersheds located in the very much different climate zones across China? This is critical, please specify clearly.

**Response**: We specified how dry and wet seasons are divided in the revised manuscript in lines 124-128 and Table 1 on page 34. In this study, the dry and wet seasons are defined according to the long-term mean monthly precipitation within a hydrological year. For watersheds in subtropical monsoon climate (Pingjiang and Xiangshui), a wet season with more than 70% of annual precipitation falling in watersheds covers a period from March to August, while the dry season starts from September to February. In other watersheds from the alpine, temperate monsoon, and temperate continental climate zones, the wet season (also called rainy season mostly in summer) starts from May to October, while dry season is from November to April.

(7) Line 117-118: Please give the PET formula.

**Response**: Revised. Please see lines 147-148.

(8) Line 121-123: How to reclassify land cover types? What is the basis for this?

**Response**: We stated how we reclassify the land cover types in the revised manuscript in lines 151-159. There are 17 types of land covers in MODIS MCD12Q1, including evergreen needleleaf forests, deciduous

needleleaf forests, evergreen broadleaf forests, deciduous broadleaf forests, mixed forests, closed shrublands, opened shrublands, woody savannas, savannas, grasslands, permanent wetlands, croplands, urban and built-up land, cropland/natural vegetation mosaics, permanent snow and ice, barren, and water bodies. We reclassify them into forest (evergreen needleleaf forests, deciduous needleleaf forests, evergreen broadleaf forests, deciduous broadleaf forests and mixed forests), shrubland (closed shrublands and opened shrublands), grassland (woody savannas, savannas and grasslands), agricultural (croplands and cropland/natural vegetation mosaics), snow (permanent snow and ice) and other lands (permanent wetlands, urban and built-up land, barren, and water bodies). Vegetation coverage including forest, shrubland and grassland can be then calculated.

(9) Line 124-134: Have you compared the two RS products with observations in the field? Which is more accurate for your study?

**Response**: We understand that it will be helpful if the accuracy of the two RS products can be validated by field observations from our study watersheds. However, it is very challenging and expensive to conduct field observations in our study watersheds that distribute across large climatic gradients. Actually, these two RS products have been widely used and have already been validated by some studies in China (Xiao et al., 2016; Yang et al., 2017). Therefore, according to previous studies, we believe the application of the two RS products can be acceptable and reliable.

In fact, the two RS products (GLASS LAI and MODIS MCD12Q1) are used for different purposes in our study. The GLASS LAI is used for quantifying watershed vegetation level that is then used to calculate ecohydrological sensitivity, while the MODIS MCD12Q1 is applied for demonstrating land use in watersheds and then used to calculate landscape indices by FRAGSTATS 4.2. Therefore, we cannot conclude which product is more accurate for our study.

[revised manuscript text omitted]